# AUTOMATED DESIGN OF AGENTIC SYSTEMS

**Shengran Hu[1,2], Cong Lu[1,2], Jeff Clune[1,2,3]**

[1]University of British Columbia, [2]Vector Institute, [3]Canada CIFAR AI Chair

{srhu,conglu}@cs.ubc.ca, jclune@gmail.com

## ABSTRACT

Researchers are investing substantial effort in developing powerful general-purpose agents, wherein Foundation Models are used as modules within *agentic systems* (e.g. Chain-of-Thought, Self-Reflection, Toolformer). However, the history of machine learning teaches us that hand-designed solutions are eventually replaced by learned solutions. We describe a newly forming research area, **A**utomated **D**esign of **A**gentic **S**ystems (**ADAS**), which aims to *automatically create powerful agentic system designs, including inventing novel building blocks and/or combining them in new ways.* We further demonstrate that there is an unexplored yet promising approach within ADAS where agents can be defined in code and new agents can be automatically discovered by a meta agent programming ever better ones in code. Given that most programming languages are Turing Complete, this approach theoretically enables the learning of *any possible* agentic system: including novel prompts, tool use, workflows, and combinations thereof. We present a simple yet effective algorithm named Meta Agent Search to demonstrate this idea, where a meta agent iteratively programs interesting new agents based on an ever-growing archive of previous discoveries. Through extensive experiments across multiple domains including coding, science, and math, we show that our algorithm can progressively invent agents with novel designs that greatly outperform state-of-the-art hand-designed agents. Importantly, we consistently observe the surprising result that agents invented by Meta Agent Search maintain superior performance even when transferred across domains and models, demonstrating their robustness and generality. Provided we develop it safely, our work illustrates the potential of an exciting new research direction toward automatically designing ever-more powerful agentic systems to benefit humanity. All code is open-sourced at https://github.com/ShengranHu/ADAS.

## 1 INTRODUCTION

Foundation Models (FMs) such as GPT (OpenAI, 2024; 2022) and Claude (Anthropic, 2024b) are quickly being adopted as powerful general-purpose agents for agentic tasks that need flexible reasoning and planning (Wang et al., 2024). Despite recent advancements in FMs, solving problems reliably often requires an agent to be a compound agentic system with multiple components instead of a monolithic model query (Zaharia et al., 2024; Rocktäschel, 2024). Additionally, to enable agents to solve complex real-world tasks, they often need access to external tools such as search engines, code execution, and database queries. As a result, many effective building blocks of agentic systems have been proposed, such as chain-of-thought planning and reasoning (Wei et al., 2022; Yao et al., 2023; Hu & Clune, 2024), memory structures (Zhang et al., 2024c; Lewis et al., 2020), tool use (Schick et al., 2023; Qu et al., 2024), and self-reflection (Madaan et al., 2024; Shinn et al., 2023). Although these agents have already seen significant success across various applications (Wang et al., 2024), developing these building blocks and combining them into complex agentic systems often requires domain-specific manual tuning and substantial effort from both researchers and engineers.

However, the history of machine learning reveals a recurring theme: manually created artifacts become replaced by learned, more efficient solutions (Clune, 2019) over time as we get more compute and data (Sutton, 2019). An early example is from computer vision, where hand-designed features like HOG (Dalal & Triggs, 2005) were eventually replaced by learned features from Convolutional Neural Networks (CNNs, Krizhevsky et al. (2012)). More recently, AutoML methods (Hutter et al.,

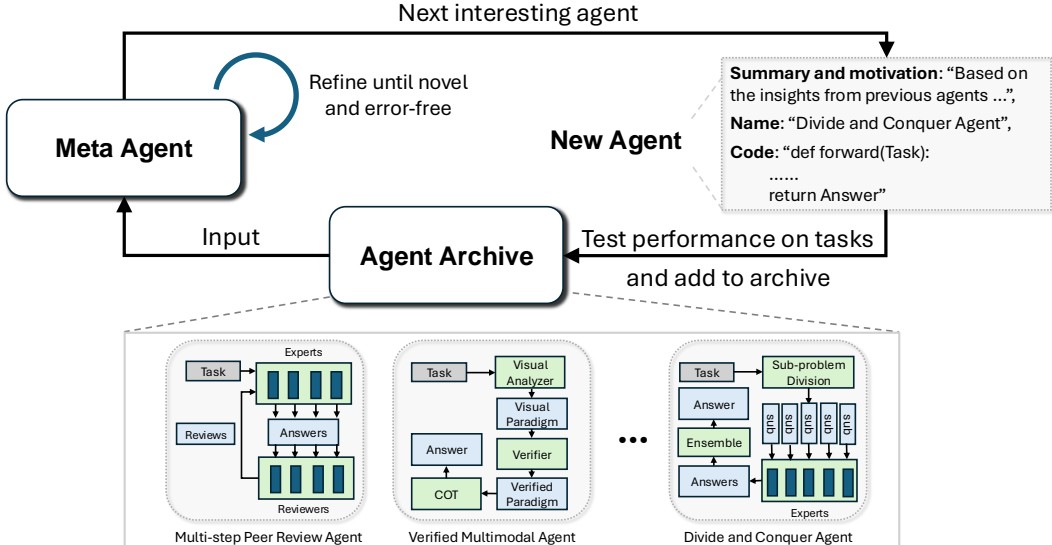

Figure 1: **Overview of the proposed algorithm Meta Agent Search and examples of discovered agents.** In our algorithm, we instruct the "meta" agent to iteratively program new agents, test their performance on tasks, add them to an archive of discovered agents, and use this archive to inform the meta agent in subsequent iterations. We show three example agents across our runs, with all names generated by the meta agent. The detailed code of example agents can be found in Appendix H.

2019) and AI-Generating Algorithms (AI-GAs, Clune (2019)) have also demonstrated the superiority of learned AI systems compared to hand-designed AI systems. For example, the current best-performing CNN models come from Neural Architecture Search (Elsken et al., 2019; Shen et al., 2023) instead of manual design; in LLM alignment, learned loss functions (Lu et al., 2024a) outperform most hand-designed ones such as DPO (Rafailov et al., 2024); The AI Scientist (Lu et al., 2024b) demonstrates an automated research pipeline, including the development of novel ML algorithms; and an endless number of robotics learning environments can be automatically generated in works like OMNI-EPIC (Faldor et al., 2024), which demonstrate surprising creativity in generated environments and allow more efficient environment creation than the manual approach (see more examples in Section 5). Therefore, in this paper, we propose a new research question: *Can we automate the design of agentic systems?*

To explore the above research question, we describe a newly forming research area we call **A**utomated **D**esign of **A**gentic **S**ystems (**ADAS**), which aims to automatically invent novel building blocks and design powerful agentic systems (Section 2). We argue that ADAS may prove to be the fastest path to developing powerful agents, and show initial evidence that learned agents can greatly outperform hand-designed agents. Considering the tremendous number of building blocks yet to be discovered in agentic systems (Section 5), it would take a long time for our research community to discover them all. Even if we successfully discover most of the useful building blocks, combining them into effective agentic systems for massive real-world applications would still be challenging and time-consuming, given the many different ways the building blocks can combine and interact with each other. In contrast, with ADAS, the building blocks and agents can be learned in an automated fashion. ADAS may not only potentially save human effort in developing powerful agents but also could be a faster path to more effective solutions than manual design.

Although a few existing works can be considered as ADAS methods, most of them focus only on designing prompts (Yang et al., 2024; Fernando et al., 2024), greatly limiting their ability to invent flexible design patterns in agents (Section 5). In this paper, we show that there is an unexplored yet promising approach to ADAS where we can define the entire agentic system in code and new agents can be automatically discovered by a "meta" agent programming ever better ones in code. Given that most programming languages, such as Python, which we use in this paper, are Turing Complete (Boyer & Moore, 1983; Ladha, 2024), searching within a code space theoretically enables an ADAS algorithm to discover *any* possible agentic systems, including all components such as

prompts, tool use, workflows, and more. Furthermore, with recent FMs being increasingly proficient in coding, we can use FMs as meta agents to create new agents in code for ADAS, enabling novel agents to be programmed in an automated manner.

Following the aforementioned ideas, we present Meta Agent Search in this paper as one of the first algorithms in ADAS that enables complete design in code space (Figure 1). The core concept of Meta Agent Search is to instruct a meta agent to iteratively create interestingly new agents, evaluate them, add them to an archive that stores discovered agents, and use this archive to help the meta agent in subsequent iterations create yet more interestingly new agents. Similar to existing open-endedness algorithms that leverage human notions of interestingness (Zhang et al., 2024a; Lu et al., 2024c), we encourage the meta agent to explore interesting (e.g., novel or worthwhile) agents. To validate the proposed approach, we evaluate the proposed Meta Agent Search on: (1) the challenging ARC logic puzzle task (Chollet, 2019) that aims to test the general intelligence of an AI system, (2) four popular benchmarks on reading comprehension, math, science questions, and multi-task problem solving, and (3) the transferability of discovered agents to held-out domains and models (Section 4).

Our experiments show that the discovered agents substantially outperform state-of-the-art hand-designed baselines. For instance, our agents improve F1 scores on reading comprehension tasks in DROP (Dua et al., 2019) by **13.6**/100 and accuracy rates on math tasks in MGSM (Shi et al., 2023) by **14.4%**. Additionally, they improve accuracy over baselines by **25.9%** and **13.2%** on GSM8K (Cobbe et al., 2021) and GSM-Hard (Gao et al., 2023) math tasks, respectively, *after transferring* across domains. The promising performance of our algorithm over hand-designed solutions illustrates the potential of ADAS in automating the design of agentic systems. Furthermore, the experiments demonstrate that the discovered agents not only perform well when transferring across similar domains but also exhibit strong performance when transferring across dissimilar domains, such as from mathematics to reading comprehension. This highlights the robustness and transferability of the agentic systems discovered by Meta Agent Search. In conclusion, our work opens up many exciting research directions and encourages further studies (Section 6).

## 2 AUTOMATED DESIGN OF AGENTIC SYSTEMS (ADAS)

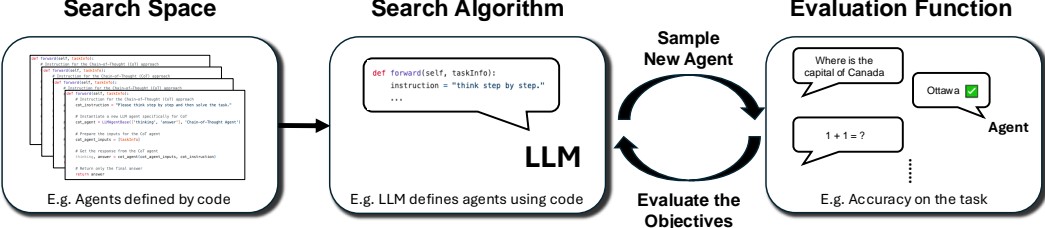

Figure 2: **The three key components of Automated Design of Agentic Systems (ADAS).** The search space determines which agentic systems can be represented in ADAS. The search algorithm specifies how the ADAS method explores the search space. The evaluation function defines how to evaluate a candidate agent on target objectives such as performance.

At the time of writing, the community has not reached a consensus on the definitions or terminologies of agents. Here, by agents we refer to agentic systems that involve Foundation Models (FMs) as modules in the workflow to solve tasks by planning, using tools, and carrying out multiple, iterative steps of processing (Chase, 2024; Ng, 2024). In this paper, we describe a newly forming research area Automated Design of Agentic Systems (ADAS). Similar to research areas in AI-GAs (Clune, 2019) and AutoML (Hutter et al., 2019), such as Neural Architecture Search (Elsken et al., 2019), we formulate ADAS as an optimization process and identify three key components of ADAS algorithms (Figure 2).

> **Formulation**
>
> Automated Design of Agentic Systems (ADAS) involves using a **search algorithm** to discover agentic systems across a **search space** that **optimize** an **evaluation function**.

- **Search Space**: The search space defines which agentic systems can be represented and thus discovered in ADAS. For example, works like PromptBreeder (Fernando et al., 2024) mutate only the text prompts of an agent, but their other components, such as workflow, remain the same. Thus, in these search spaces, agents that have a different workflow than the predefined one can not be represented. Existing works also explore search spaces such as graph structures (Zhuge et al., 2024) and feed-forward networks (Liu et al., 2023).

- **Search Algorithm**: The search algorithm defines how ADAS algorithms explore the search space. Since the search space is often very large or even unbounded, the exploration-exploitation trade-off (Sutton & Barto, 2018) should be considered. Ideally, the algorithm can both quickly discover high-performance agentic systems and avoid remaining stuck in a local optimum. Existing approaches include using Reinforcement Learning (Zhuge et al., 2024) or an FM iteratively generating new solutions (Fernando et al., 2024) as search algorithms.

- **Evaluation Function**: Depending on the application of the ADAS algorithm, we may consider different objectives to optimize, such as performance, cost, latency, or safety of agents. An evaluation function defines how to evaluate a candidate agent on those objectives. For example, to assess the agent's performance on unseen future data, a simple method is to calculate the accuracy rate on the validation data for a task, which is commonly adopted in existing works (Zhuge et al., 2024; Fernando et al., 2024).

Although many search space designs are possible and some have already been explored (Section 5), there is an unexplored yet promising approach where we can define the entire agentic system in code and new agents can be automatically discovered by a meta agent programming ever better ones in code. Searching within a code space theoretically enables the ADAS algorithm to discover *any* possible building blocks (e.g., prompts, tool use, workflow) and agentic systems that combine any of these building blocks in any way. This approach also offers better interpretability for agent design patterns since the program code is often readable, making debugging easier and enhancing AI safety. Additionally, compared to search spaces using networks (Liu et al., 2023) or graphs (Zhuge et al., 2024), searching in a code space allows us to more easily build on existing human efforts. For example, it is possible to search within open-source agent frameworks like LangChain (LangChainAI, 2022) and build upon all existing building blocks (e.g., RAG, search engine tools). Finally, since FMs are proficient in coding, utilizing a code search space allows us to leverage existing expertise from FMs during the search process. In contrast, search algorithms in custom search spaces, such as graphs, may be much less efficient due to the absence of these priors. Therefore, we argue that the approach of using programming languages as the search space should be studied more in ADAS.

## 3 OUR ALGORITHM: META AGENT SEARCH

In this section, we present Meta Agent Search, a simple yet effective algorithm to demonstrate the approach of defining and searching for agents in code. The core idea of Meta Agent Search is to adopt FMs as meta agents to iteratively program interestingly new agents based on an ever-growing archive of previous discoveries. Although any possible building blocks and agentic systems can theoretically be programmed by the meta agent from scratch, it is inefficient in practice to avoid providing the meta agent any basic functions such as FM query APIs or existing tools. Therefore, in this paper, we define a simple framework (within 100 lines of code) for the meta agent, providing it with a basic set of essential functions like querying FMs or formatting prompts. As a result, the meta agent only needs to program a "forward" function to define a new agentic system, similar to the practice in FunSearch (Romera-Paredes et al., 2024). This function takes in the information of the task and outputs the agent's response to the task. Details of the framework codes and examples of the agents defined with this framework can be found in Appendix D.

As shown in Figure 1, the core idea of Meta Agent Search is to have a meta agent iteratively program new agents in code. The algorithm proceeds as follows: (1) The archive is (optionally) initialized with baseline agents such as Chain-of-Thought (Wei et al., 2022) and Self-Refine (Madaan et al., 2024; Shinn et al., 2023). (2) Conditioned on the archive, the meta agent designs a new agent by generating a high-level description of the new idea for an agentic system and then implementing it in code. The design then undergoes two self-reflection (Madaan et al., 2024; Shinn et al., 2023) steps by the meta agent to ensure it is novel. (3) The generated agent is evaluated using validation data from the target domain. If errors occur during evaluation, the meta agent performs a self-reflection step to

refine the design, repeating this process up to five times if necessary. (4) Finally, the agent is added to the archive along with its evaluation metrics, and the process continues with the updated archive until the maximum number of iterations is reached. A pseudocode of the algorithm is provided in Appendix I.

Similar to existing open-endedness algorithms that leverage human notions of interestingness (Zhang et al., 2024a; Lu et al., 2024c), we encourage the meta agent to explore interestingly new (e.g., novel or worthwhile) agents based on an ever-growing archive of previous discoveries. Here, we calculate the performance (e.g., success rate or F1 score) as the metrics for the meta agent to maximize. The prompt and more details are presented in Appendix C.

## 4 EXPERIMENTS

We conduct extensive experiments on: (1) the ARC challenge (Chollet, 2019) (Section 4.1), (2) four popular benchmarks assessing the agent's abilities on reading comprehension, math, science questions, and multi-task problem solving (Section 4.2), and (3) the transferability of discovered agents on math to held-out math tasks and non-math tasks (Section 4.3). We use an identical implementation of the algorithm across different tasks, with the only variation being task-specific descriptive text included in the prompt (details are available in Appendix C). Across all experiments, we find that the discovered agents substantially outperform baseline state-of-the-art hand-designed agents and maintain superior performance even when transferred across domains and models.

### 4.1 CASE STUDY: ARC CHALLENGE

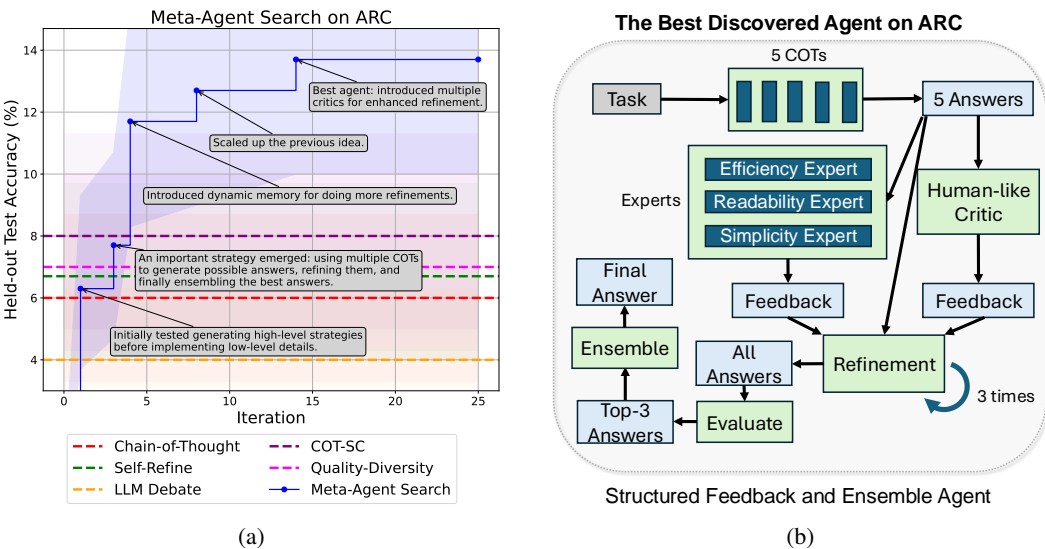

(a)                 (b)

Figure 3: **The results of Meta Agent Search on the ARC challenge.** (a) Meta Agent Search progressively discovers high-performance agents based on an ever-growing archive of previous discoveries. We report the median accuracy and the 95% bootstrap confidence interval on a held-out test set by evaluating agents five times. (b) The visualization of the best agent discovered by Meta Agent Search on the ARC challenge. Detailed implementation of this agent is available in Appendix E.

We first demonstrate how Meta Agent Search discovers novel agentic systems and outperforms existing state-of-the-art hand-designed agents in the Abstraction and Reasoning Corpus (ARC) challenge (Chollet, 2019). This challenge aims to evaluate the general intelligence of AI systems through their ability to acquire new skills. Questions in ARC include (1) showing multiple examples of visual input-output grid patterns, (2) the AI system learning the transformation rule of grid patterns from examples, and (3) predicting the output grid pattern given a test input grid pattern. Since each question in ARC has a unique transformation rule, it requires the AI system to learn efficiently with few-shot examples, leveraging capabilities in number counting, geometry, and topology.

**Setup.** Following common practice (Greenblatt, 2024), we require the agent to write code for the transformation rule instead of answering directly. We provide tool functions in the framework (described in Section 3) that evaluate the generated transformation code. Given the significant challenge that ARC poses to current AI systems, we sample our data from questions with grid dimensions $\leq 5 \times 5$ in the "Public Training Set (Easy)". We sample a validation set and a test set with 20 and 60 questions, respectively, for searching and testing. We calculate the validation and test accuracy of an agent by assessing it over the validation and test sets five times to reduce the variance from the stochastic sampling of FMs. We evaluate all discovered agents on the held-out test set and report the test accuracy in Figure 3. Meta Agent Search runs for 25 iterations and the meta agent uses GPT-4 (OpenAI, 2024), while discovered agents and baselines are evaluated using GPT-3.5 (OpenAI, 2022) to reduce compute cost. More algorithmic details and examples of ARC questions can be found in Appendix E.

**Baselines.** We compared against five state-of-the-art hand-designed agents: (1) Chain-of-Thought (COT, Wei et al. (2022)), which instructs the agent to output the reasoning before answering to improve complex problem-solving through intermediate steps; (2) Self-Consistency with Chain-of-Thought (COT-SC, Wang et al. (2023b)), which ensembles multiple parallel answers from COT to produce a more accurate answer; (3) Self-Refine (Madaan et al., 2024; Shinn et al., 2023), which allows iterative self-reflection to correct mistakes made in previous attempts; (4) LLM-Debate (Du et al., 2023), which enables different LLMs to debate with each other, leveraging diverse perspectives to find better answers; (5) Quality-Diversity, a simplified version of Intelligent Go-Explore (Lu et al., 2024c), which produces and ensembles diverse answers to better explore potential solutions. The selected baselines represent widely adopted agent designs in the agent literature, embodying key design patterns and approaches frequently utilized across various applications. By "state-of-the-art," we refer to these baseline designs as exemplifying important advancements and practices within the field. We also use all baselines as initial seeds in the archive for Meta Agent Search, with additional results for empty initialization provided in Appendix J. To ensure fair comparisons, all baseline implementations were developed using the same framework as the Meta Agent, providing a consistent and equitable evaluation environment. More details about baselines can be found in Appendix G.

**Results and Analysis.** As shown in Figure 3a, Meta Agent Search effectively and progressively discovers agents that perform better than state-of-the-art hand-designed baselines. Important breakthroughs are highlighted in the text boxes. As is critical in prior works on open-endedness and AI-GAs (Zhang et al., 2024a; Faldor et al., 2024; Wang et al., 2019; 2020; Lehman & Stanley, 2011), Meta Agent Search innovates based on a growing archive of previous stepping stones. For example, an important design pattern emerged in iteration 3 where it uses multiple COTs to generate possible answers, refines them, and finally ensembles the best answers. This became a crucial stepping stone that subsequent designs tended to utilize. Additionally, the best-discovered agent is shown in Figure 3b, where a complex feedback mechanism is adopted to refine answers more effectively. Careful observation of the search progress reveals that this sophisticated feedback mechanism did not appear suddenly. Instead, the ideas of incorporating diverse feedback, evaluating for various specific traits (via experts) such as efficiency and simplicity, and simulating human-like feedback emerged in iterations 5, 11, and 12, respectively. The final mechanism is an innovation based on these three stepping stones. This illustrates that even though these stepping stones did not achieve high performance immediately upon emergence, later discoveries benefited from these innovations by combining different stepping stones, resembling crossover in evolution via LLMs (Meyerson et al., 2023). Overall, the results showcase the potential of ADAS and the effectiveness of Meta Agent Search to progressively discover agents that outperform state-of-the-art hand-designed baselines and invent novel design patterns through the innovation and combination of stepping stones.

## 4.2 REASONING AND PROBLEM-SOLVING DOMAINS

**Setup.** Next, we investigate the potential of our algorithm to improve the capabilities of agents across math, reading, and reasoning domains. We test Meta Agent Search on four popular benchmarks: (1) DROP (Dua et al., 2019) for evaluating **Reading Comprehension**; (2) MGSM (Shi et al., 2023) for evaluating **Math** capability under a multi-lingual setting; (3) MMLU (Hendrycks et al., 2021) for evaluating **Multi-task** Problem Solving; and (4) GPQA (Rein et al., 2023) for evaluating the capability of solving hard (graduate-level) questions in **Science**. The search is conducted independently within each domain. Meta Agent Search runs for 30 iterations. The meta agent uses GPT-

4 (OpenAI, 2024), while the discovered agents and baselines are evaluated using GPT-3.5 (OpenAI, 2022). More details about datasets and experiment settings can be found in Appendix F.

**Baselines.** We adopt all baselines introduced in Section 4.1. Additionally, since the above domains require strong reasoning skills, we include two additional baselines that specifically focus on enhancing the reasoning capabilities of agents for a more thorough comparison: (1) Step-back Abstraction (Zheng et al., 2023), which instructs agents to first consider the principles involved in solving the task for better reasoning; (2) Role Assignment (Xu et al., 2023), which assigns different roles to FMs to obtain better answers. Furthermore, we compare our approach with the state-of-the-art prompt optimization baseline OPRO (Yang et al., 2024) to highlight the advantages of learning all possible components of agents rather than focusing solely on prompts. More details about the baselines can be found in Appendix G.

Table 1: **Performance comparison between Meta Agent Search and state-of-the-art hand-designed agents across multiple domains.** Meta Agent Search discovers superior agents compared to the baselines in every domain. We report the test accuracy and the 95% bootstrap confidence interval on held-out test sets. The search is conducted independently for each domain. Here, and in all tables below, we bold the entry with the highest performance for each domain, as well as all entries whose median falls within the 95% confidence interval of the highest-performing treatment.

| Agent Name | F1 Score | Accuracy (%) | | |
|---|---|---|---|---|
| | Reading Comprehension | Math | Multi-task | Science |
| *State-of-the-art Hand-designed Agents* | | | | |
| Chain-of-Thought (Wei et al., 2022) | $64.2 \pm 0.9$ | $28.0 \pm 3.1$ | $65.4 \pm 3.3$ | $29.2 \pm 3.1$ |
| COT-SC (Wang et al., 2023b) | $64.4 \pm 0.8$ | $28.2 \pm 3.1$ | $65.9 \pm 3.2$ | $30.5 \pm 3.2$ |
| Self-Refine (Madaan et al., 2024) | $59.2 \pm 0.9$ | $27.5 \pm 3.1$ | $63.5 \pm 3.4$ | $\mathbf{31.6 \pm 3.2}$ |
| LLM Debate (Du et al., 2023) | $60.6 \pm 0.9$ | $39.0 \pm 3.4$ | $65.6 \pm 3.3$ | $\mathbf{31.4 \pm 3.2}$ |
| Step-back Abstraction (Zheng et al., 2023) | $60.4 \pm 1.0$ | $31.1 \pm 3.2$ | $65.1 \pm 3.3$ | $26.9 \pm 3.0$ |
| Quality-Diversity (Lu et al., 2024c) | $61.8 \pm 0.9$ | $23.8 \pm 3.0$ | $65.1 \pm 3.3$ | $30.2 \pm 3.1$ |
| Role Assignment (Xu et al., 2023) | $65.8 \pm 0.9$ | $30.1 \pm 3.2$ | $64.5 \pm 3.3$ | $31.1 \pm 3.1$ |
| *Automated Design of Agentic Systems on Different Domains* | | | | |
| Prompt Optimization (Yang et al., 2024) | $69.1 \pm 0.9$ | $30.6 \pm 3.2$ | $\mathbf{67.6 \pm 3.2}$ | $\mathbf{32.9 \pm 3.2}$ |
| Meta Agent Search (Ours) | $\mathbf{79.4 \pm 0.8}$ | $\mathbf{53.4 \pm 3.5}$ | $\mathbf{69.6 \pm 3.2}$ | $\mathbf{34.6 \pm 3.2}$ |

**Results and Analysis.** The results across multiple domains demonstrate that Meta Agent Search can discover agents that outperform state-of-the-art hand-designed agents (Table 1). We want to highlight the substantial gap between the learned agents and hand-designed agents in the Reading Comprehension and Math domains, with improvements in F1 scores by **13.6**/100 and accuracy rates by **14.4%**, respectively. While Meta Agent Search also outperforms baselines in the Multi-task and Science domains, the gap is smaller. We hypothesize that for challenging questions in the Science and Multi-task domains, the knowledge in FMs is not sufficient to solve the questions, limiting the improvement through optimizing agentic systems, which is a problem that will diminish as FMs improve. In contrast, in the Reading Comprehension and Math domains, FMs possess adequate knowledge to solve the questions, and errors could mainly be hallucinations or calculation mistakes, which can be mitigated through well-designed agentic systems, like the ones discovered by Meta Agent Search. Additionally, when compared to prompt optimization methods, the results demonstrate that our proposed Meta Agent Search consistently outperforms them across all domains. This comparison further strengthens our argument that defining agents in code and enabling the learning of all components offer significant advantages. Overall, the results across various domains showcase the effectiveness of Meta Agent Search in searching for agents tailored to specific domains. This could be increasingly useful for saving human efforts and developing better task-specific agents as we continue to create agents for a diverse set of applications (Wang et al., 2024).

### 4.3 GENERALIZATION AND TRANSFERABILITY

In the previous sections, we illustrated that Meta Agent Search can find effective agents for individual tasks. In this section, we further demonstrate the transferability and generalizability of the discovered agents. To demonstrate the generalizability of the invented building blocks and de-

sign patterns, we transfer discovered agents from the MGSM (Math) domain to both math and non-math domains to test their ability to generalize across different tasks. We evaluate the top 3 agents from MGSM by transferring them to (1) popular math domains: GSM8K (Cobbe et al., 2021), GSM-Hard (Gao et al., 2023), and (2) non-math domains: MMLU (Multi-task) and DROP (Reading Comprehension), as detailed in Section 4.2. As shown in Table 2, Meta Agent Search consistently outperforms the baselines. Notably, our agents improve accuracy by **25.9%** on GSM8K and **13.2%** on GSM-Hard compared to the baselines when transferring within math domains. More surprisingly, we find that agents discovered in the math domain can also be transferred to non-math domains. While their performance does not fully match agents specifically designed for the target domains, they still outperform state-of-the-art hand-designed baselines. More results of transfers across domains are shown in Appendix B.

We also observe similar superiority when transferring agents across different FMs on ARC. We test the top 3 agents with the best test accuracy evaluated with GPT-3.5 on ARC and then transfer them to Claude-Haiku (Anthropic, 2024a), GPT-4 (OpenAI, 2024), and Claude-Sonnet (Anthropic, 2024b). As shown in Table 3, we observe that the searched agents consistently outperform the hand-designed agents, with a substantial gap. Notably, we found that Claude-Sonnet, the most powerful model from Anthropic, performs the best among all tested models, enabling our best agent to achieve nearly **50%** accuracy on ARC. These results on transferring across domains and models highlight Meta Agent Search 's ability to discover generalizable design patterns and agentic systems.

Table 2: **Performance on held-out math and non-math domains when transferring top agents from MGSM (Math).** GSM8K and GSM-Hard are the held-out math domains, while MMLU is for Multi-task, and DROP is for Reading Comprehension. Agents discovered by Meta Agent Search consistently outperform the baselines across all domains. We report the test accuracy and the 95% bootstrap confidence interval. The names of the top agents are generated by Meta Agent Search.

| Agent Name | Accuracy (%) | | | | F1 Score |
| --- | --- | --- | --- | --- | --- |
| | MGSM | GSM8K | GSM-Hard | MMLU | DROP |
| **Manually Designed Agents** | | | | | |
| Chain-of-Thought (Wei et al., 2022) | $28.0 \pm 3.1$ | $34.9 \pm 3.2$ | $15.0 \pm 2.5$ | $\mathbf{65.4 \pm 3.3}$ | $64.2 \pm 0.9$ |
| COT-SC (Wang et al., 2023b) | $28.2 \pm 3.1$ | $37.8 \pm 3.4$ | $15.5 \pm 2.5$ | $\mathbf{65.9 \pm 3.2}$ | $64.4 \pm 0.8$ |
| Self-Refine (Madaan et al., 2024) | $27.5 \pm 3.1$ | $38.9 \pm 3.4$ | $15.1 \pm 2.4$ | $63.5 \pm 3.4$ | $59.2 \pm 0.9$ |
| LLM Debate (Du et al., 2023) | $39.0 \pm 3.4$ | $43.6 \pm 3.4$ | $17.4 \pm 2.6$ | $\mathbf{65.6 \pm 3.3}$ | $60.6 \pm 0.9$ |
| Step-back Abstraction (Zheng et al., 2023) | $31.1 \pm 3.2$ | $31.5 \pm 3.3$ | $12.2 \pm 2.3$ | $\mathbf{65.1 \pm 3.3}$ | $60.4 \pm 1.0$ |
| Quality-Diversity (Lu et al., 2024c) | $23.8 \pm 3.0$ | $28.0 \pm 3.1$ | $14.1 \pm 2.4$ | $\mathbf{65.1 \pm 3.1}$ | $61.8 \pm 0.9$ |
| Role Assignment (Xu et al., 2023) | $30.1 \pm 3.2$ | $37.0 \pm 3.4$ | $18.0 \pm 2.7$ | $64.5 \pm 3.3$ | $65.8 \pm 0.9$ |
| **Top Agents Searched on MGSM (Math)** | | **Transferred within Math Domains** | | **Transferred beyond Math Domains** | |
| Dynamic Role-Playing Architecture | $\mathbf{53.4 \pm 3.5}$ | $\mathbf{69.5 \pm 3.2}$ | $\mathbf{31.2 \pm 3.2}$ | $62.4 \pm 3.4$ | $70.4 \pm 0.9$ |
| Structured Multimodal Feedback Loop | $\mathbf{50.2 \pm 3.5}$ | $64.5 \pm 3.4$ | $\mathbf{30.1 \pm 3.2}$ | $\mathbf{67.0 \pm 3.2}$ | $70.4 \pm 0.9$ |
| Interactive Multimodal Feedback Loop | $47.4 \pm 3.5$ | $64.9 \pm 3.3$ | $27.6 \pm 3.2$ | $\mathbf{64.8 \pm 3.3}$ | $\mathbf{71.9 \pm 0.8}$ |

## 5 RELATED WORK

**Agentic Systems.** Researchers develop various building blocks and design patterns for different applications. Important building blocks for agentic systems include: prompting techniques (Chen et al., 2023a; Schulhoff et al., 2024), chain-of-thought-based planning and reasoning methods (Wei et al., 2022; Yao et al., 2023; Hu & Clune, 2024), reflection (Madaan et al., 2024; Shinn et al., 2023), developing new skills for embodied agents in code (Wang et al., 2023a; Vemprala et al., 2023), external memory and RAG (Zhang et al., 2024c; Lewis et al., 2020), tool use (Qu et al., 2024; Schick et al., 2023; Nakano et al., 2021), assigning FM modules in the agentic system with different roles and enabling them to collaborate (Hong et al., 2023; Wu et al., 2023; Qian et al., 2023; Xu et al., 2023; Qian et al., 2024), and enabling the agent to instruct itself for the next action (Richards, 2023), etc. While the community has invested substantial effort in developing all the above important techniques, this is only a partial list of the discovered building blocks, and many more remain to be

Table 3: **Performance on ARC when transferring top agents from GPT-3.5 to other FMs.** Agents discovered by Meta Agent Search consistently outperform the baselines across different models. We report the test accuracy and the 95% bootstrap confidence interval. The names of top agents are generated by Meta Agent Search. [†]We manually changed this name because the original generated name was confusing.

| Agent Name | Accuracy on ARC (%) | | | |
|---|---|---|---|---|
| | **GPT-3.5** | **Claude-Haiku** | **GPT-4** | **Claude-Sonnet** |
| **Manually Designed Agents** | | | | |
| Chain-of-Thought (Wei et al., 2022) | $6.0 \pm 2.7$ | $4.3 \pm 2.2$ | $17.7 \pm 4.4$ | $25.3 \pm 5.0$ |
| COT-SC (Wang et al., 2023b) | $8.0 \pm 3.2$ | $5.3 \pm 2.5$ | $19.7 \pm 4.5$ | $26.3 \pm 4.9$ |
| LLM Debate (Du et al., 2023) | $4.0 \pm 2.2$ | $1.7 \pm 1.5$ | $19.0 \pm 4.5$ | $24.7 \pm 4.8$ |
| Self-Refine (Madaan et al., 2024) | $6.7 \pm 2.7$ | $\mathbf{6.3 \pm 2.8}$ | $23.0 \pm 5.2$ | $\mathbf{39.3 \pm 5.5}$ |
| Quality-Diversity (Lu et al., 2024c) | $7.0 \pm 2.9$ | $3.3 \pm 2.2$ | $23.0 \pm 4.7$ | $31.7 \pm 5.3$ |
| **Top Agents Searched with GPT-3.5** | | **Transferred to Other FMs** | | |
| Structured Feedback and Ensemble Agent | $\mathbf{13.7 \pm 3.9}$ | $5.0 \pm 2.5$ | $30.0 \pm 5.2$ | $38.7 \pm 5.5$ |
| Hierarchical Committee Reinforcement Agent | $\mathbf{13.3 \pm 3.8}$ | $\mathbf{8.3 \pm 3.2}$ | $\mathbf{32.3 \pm 8.9}$ | $39.7 \pm 5.5$ |
| Dynamic Memory and Refinement Agent[†] | $\mathbf{12.7 \pm 3.9}$ | $\mathbf{9.7 \pm 3.3}$ | $\mathbf{37.0 \pm 5.3}$ | $\mathbf{48.3 \pm 5.7}$ |

uncovered. Therefore, in this paper, we describe a newly forming research area, ADAS, which aims to invent novel building blocks and design powerful agentic systems in an automated manner.

**Existing Attempts to ADAS.** There are two categories of works that attempt ADAS: those focused on learning better prompts and those that learn more components beyond prompts. Most works fall into the first category, where FMs are used to automate prompt engineering, primarily enhancing the phrasing of instructions to improve reasoning (Yang et al., 2024; Fernando et al., 2024; Zhou et al., 2024a; Yuksekgonul et al., 2024). However, these prompts are often domain-specific and difficult to generalize. Some works optimize role definitions within prompts (Yuan et al., 2024; Chen et al., 2023c;b; Wu et al., 2023), as assigning personas or roles to agents has been shown to be beneficial (Xu et al., 2023). Although tuning prompts can improve performance, other components remain fixed, limiting the space of agents that can be discovered. The second category, which is less explored, involves learning additional components such as workflows, often representing agents as networks or graphs. In these formulations, the FM with a certain prompt is considered a transformation function for text on nodes, and the information flow of the text is considered as edges. For example, DyLAN (Liu et al., 2023) uses FMs to optimize connections between nodes in a network, DSPy (Khattab et al., 2024) and Trace (Cheng et al., 2024) optimizes across the Cartesian product of a set of possible nodes, and GPT-Swarm (Zhuge et al., 2024) uses reinforcement learning to optimize node connections. Although these approaches optimize workflows, many components like tool usage remain fixed. AgentOptimizer (Zhang et al., 2024b) learns the tools used in agents, AutoFlow (Li et al., 2024) proposes a new language to optimize workflow, Agent Symbolic Learning (Zhou et al., 2024b) attempts to learn prompts, tools, and workflows together. While these works share similar motivations to learn more components in agents, they either fail to cover all possible designs in agentic systems or have harder search spaces for search algorithms. In contrast, our work represents all components in code, allowing all possible designs in agentic systems and resulting in a promising search space for FM-guided search, as coding tasks are one of the most important tasks in FMs' training. We also include additional related work in Appendix A.1.

## 6 DISCUSSION AND CONCLUSION

**Safety Considerations.** While it is highly unlikely that model-generated code will perform overtly malicious actions in our current settings with the Foundation Models (FMs) we employ, such code could still act destructively due to limitations in model capability or alignment (Rokon et al., 2020; Chen et al., 2021). To address these risks, we have implemented safety measures including containerized execution of all generated code in secure, isolated environments, thorough manual inspections to verify the absence of harmful behaviors, and clear warnings in our codebase to alert users to potential risks. These practices align with established safety standards in the literature, such

as those in SWE-Bench (Jimenez et al., 2024) and Voyager (Wang et al., 2023a), which similarly prioritize controlled execution environments.

The proposed Automated Design of Agentic Systems (ADAS) introduces a novel area in AI-GA research, potentially accelerating the development of Artificial General Intelligence (AGI) beyond current manual approaches (Clune, 2019). This raises broader questions about advancing AI capabilities, a topic extensively debated in prior works (Clune, 2019; Ecoffet et al., 2020; Bostrom, 2002; Yudkowsky et al., 2008; Bengio et al., 2024), though beyond this paper's scope. We argue that publishing this work is net beneficial. It reveals that powerful ADAS algorithms can be easily programmed using API access to FMs, without requiring expensive hardware like GPUs, informing the community of their accessibility and implications. Moreover, ADAS can enhance safety in agentic systems by automating the design of explicit, interpretable workflows, reducing the risk of malicious behavior through greater controllability and auditability.

We believe the discussion on ADAS and its safety impact is timely given the growing adoption of agentic systems in real-world applications (Turow, 2024), where ADAS can streamline the creation of safe, reliable agents, amplifying AI's potential to benefit humanity in domains like health and economics (Amodei, 2024). Furthermore, as self-improving AI systems become prominent (Clune, 2019; Fernando et al., 2024; Lu et al., 2024a; Zelikman et al., 2022), their continued development appears inevitable. By sharing this work, we aim to inspire further research into safe-ADAS algorithms—potentially incorporating mechanisms like Constitutional AI (Bai et al., 2022)—to ensure that advancements in AI-GA and self-improving AI yield systems that are both powerful and aligned with human values, ultimately fostering safer AI development.

**Future Work.** Our work also opens up many future research directions. Below, we discuss a few, with additional directions provided in Appendix A.2.

- **Higher-order ADAS.** Since the meta agent used in ADAS to program new agents in code is also an agent, ADAS can become self-referential where the meta agent can be improved through ADAS as well. It would be an exciting direction to have a higher order of meta-learning to allow the learning of the meta agent and even the meta-meta agent, etc. (Lu et al., 2023; Schmidhuber, 1987; 2003; Zelikman et al., 2024)

- **Online Continual Learning.** As agents are deployed, they will receive vast amounts of feedback from both task environments and users. Continuously improving agents based on this extensive feedback is challenging for human developers. However, with ADAS automating the design and enhancement of agents, online continual learning becomes feasible post-deployment.

- **Multi-objective ADAS.** We only consider one objective (i.e., performance) to optimize in this paper, but in practice, multiple objectives are often considered, such as cost, latency, and robustness of agentic systems (Hu et al., 2021; Huang et al., 2023). Thus, integrating multi-objective search algorithms (Deb et al., 2002) in ADAS could be promising.

- **Towards a Better Understanding of FMs.** Works from Neural Architecture Search (Huang et al., 2023) show that by observing the emerged architecture, we could gain more insights into Neural Networks. In this paper, we also gained insights about FMs from the results. For example, the best agent with GPT-3.5 involves a complex feedback mechanism, but when we transfer to other advanced models, the agent with a simpler feedback mechanism but more refinement becomes a better agent (Section 4.3). This shows that GPT-3.5 may have a worse capability in evaluating and refining the answers, so it needs a complex feedback mechanism for better refinement, while other advanced models benefit more from a simpler feedback mechanism.

**Conclusion.** In this paper, we propose a new research problem, Automated Design of Agentic Systems (ADAS), which aims to *automatically invent novel building blocks and design powerful agentic systems*. We demonstrated that a promising approach to ADAS is to define agents in code, allowing new agents to be automatically discovered by a "meta" agent programming them in code. Following this idea, we propose Meta Agent Search, where the meta agent iteratively builds on previous discoveries to program interesting new agents. The experiments show that Meta Agent Search consistently outperforms state-of-the-art hand-designed agents across an extensive number of domains, and the discovered agents transfer well across models and domains. Overall, our work illustrates the potential of an exciting new research direction toward full automation in developing powerful agentic systems from the bottom up.

ACKNOWLEDGMENTS

This work was supported by the Vector Institute, the Canada CIFAR AI Chairs program, grants from Schmidt Futures and Open Philanthropy, an NSERC Discovery Grant, and a generous donation from Rafael Cosman. We thank Jenny Zhang, Rach Pradhan, Ruiyu Gou, Nicholas Ioannidis, and Eunjeong Hwang for insightful discussions and feedback.

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

# SUPPLEMENTARY MATERIAL

## TABLE OF CONTENTS

## A MORE RELATED WORK AND FUTURE WORK

### A.1 MORE RELATED WORK

**AI-Generating Algorithms and AutoML.** Research in AI-Generating Algorithms (AI-GAs, Clune (2019)) and AutoML (Hutter et al., 2019) aims to replace handcrafted components in AI systems by learning them. This field has three key pillars: (1) meta-learning architectures, (2) meta-learning learning algorithms, and (3) generating learning environments and training data (Clune, 2019). Neural Architecture Search (Elsken et al., 2019; Lu et al., 2019; Hu et al., 2021) exemplifies the first pillar by automating neural network design, while works like MAML (Finn et al., 2017) and Meta-RL (Wang et al., 2016; Duan et al., 2017; Norman & Clune, 2023; Zintgraf et al., 2021a;b) exemplify the second pillar, focusing on "learning to learn" for improved sample efficiency and generalizability. The third pillar includes works like POET (Wang et al., 2019; Dharna et al., 2022; Wang et al., 2020) and OMNI-EPIC (Faldor et al., 2024), which generate learning environments in an open-ended manner. We position Automated Design of Agentic Systems in both the first and second pillars: meta-learning agentic architectures and leveraging in-context learning to "learn to learn," as shown in the ARC challenge (Section 4.1). Furthermore, recent AI-GA and AutoML advances have also integrated Foundation Models (FMs) to write code, as seen in Fun-Search (Romera-Paredes et al., 2024) and EoH (Liu et al., 2024), where FMs discover optimization algorithms. In DiscoPOP (Lu et al., 2024a), FMs program loss functions for preference learning, and Eureka (Ma et al., 2023) and language-to-reward (Yu et al., 2023) enable FMs to write reward functions for reinforcement learning. OMNI-EPIC (Faldor et al., 2024) allows FMs to create robotics learning environments. Similarly, we enable FMs to program new agents in code.

### A.2 MORE FUTURE WORK

- **More complex domains.** Currently, we only evaluate Meta Agent Search on single-step QA tasks in this paper. It would be interesting to extend the method to more complex domains, such as real-world applications involving multi-step interaction with complex environments.

- **Seeding ADAS with more existing building blocks.** Although we can theoretically allow any components in agentic systems to be programmed from scratch in the code space, it is not efficient in practice. Therefore, it would be interesting to explore ADAS by standing on the shoulders of existing human efforts, such as search engine tools, RAG (Lewis et al., 2020), or functions from existing agent frameworks like LangChain (LangChainAI, 2022). Additionally, it is interesting to support multi-modal capabilities (e.g. vision) in FMs or allow different FMs to be available in agentic systems. This will enable the meta agent to choose from different FMs flexibly according to the difficulty of the instruction and whether data privacy is a priority.

- **Novelty search algorithms.** In Meta Agent Search, the design of the search algorithm is relatively simple, focusing solely on exploring interesting new designs. A more careful design of the search algorithm can be a promising future direction. For example, one could incorporate more sophisticated ideas from Quality-Diversity (Mouret & Clune, 2015; Cully & Demiris, 2017), AI-generating (Clune, 2019), and Open-ended Algorithms (Faldor et al., 2024; Zhang et al., 2024a; Stanley & Lehman, 2015; Stanley et al., 2019). One could also include more classic approaches to balance exploration and exploitation (Sutton & Barto, 2018; Liu et al., 2024).

- **More Intelligent Evaluation Functions.** In this work, we simply evaluate discovered agents on the evaluation set and use the numerical performance results. However, this approach is both expensive and misses a lot of information. A promising future direction is to enable the meta agent to analyze detailed running logs during the evaluation, which contain rich information on the failure and success modes for better debugging and improving agentic systems (Zhou et al., 2024b). Also, many tasks involve subjective answer evaluations (Chiang et al., 2024; Lu et al., 2024b) that do not have ground-truth answers. It is also important to design novel evaluation functions in ADAS to address these tasks. Finally, in this work, we targeted only one domain during the search. It would be interesting to explore whether ADAS algorithms can design even better generalist agents when specifically searching for agents capable of performing well across multiple domains.

- **Understanding the emergence of complexity from human organizations.** Beyond potentially saving researchers' efforts and improving upon the manual design of agentic systems, the research

in ADAS is also scientifically intriguing as it sheds light on the origins of complexity emerging from human organization and society. The agentic system is a machine learning system that operates primarily over natural language—a representation that is interpretable to humans and used by humans in constructing our organization and society. Thus, there is a close connection between agentic systems and human organizations, as shown in works incorporating the organizational structure for human companies in agents (Hong et al., 2023) or simulating a human town with agents (Park et al., 2023). Therefore, the study in ADAS may enable us to observe how to create a simple set of conditions and have an algorithm to bootstrap itself from simplicity to produce complexity in a system akin to human society.

## B GENERALIZATION AND TRANSFERABILITY

In this section, we present more details of the experiments in Section 4.3 and the complete results of transferring agents across different domains.

For the results shown in Table 3, we use "gpt-4o-2024-05-13" for GPT-4, "claude-3-haiku-20240307" for Claude-Haiku, and "claude-3-5-sonnet-20240620" for Claude-Sonnet.

Table 4: **Performance on different math domains when transferring top agents from MGSM to other math domains.** Agents discovered by Meta Agent Search consistently outperform the baselines across different math domains. We report the test accuracy and the 95% bootstrap confidence interval. The names of top agents are generated by Meta Agent Search.

| Agent Name | Accuracy (%) | | | | |
| --- | --- | --- | --- | --- | --- |
| | MGSM | GSM8K | GSM-Hard | SVAMP | ASDiv |
| **Manually Designed Agents** | | | | | |
| Chain-of-Thought (Wei et al., 2022) | $28.0 \pm 3.1$ | $34.9 \pm 3.2$ | $15.0 \pm 2.5$ | $77.8 \pm 2.8$ | $88.9 \pm 2.2$ |
| COT-SC (Wang et al., 2023b) | $28.2 \pm 3.1$ | $37.8 \pm 3.4$ | $15.5 \pm 2.5$ | $78.2 \pm 2.8$ | $89.0 \pm 2.1$ |
| Self-Refine (Madaan et al., 2024) | $27.5 \pm 3.1$ | $38.9 \pm 3.4$ | $15.1 \pm 2.4$ | $\mathbf{78.5 \pm 2.8}$ | $\mathbf{89.2 \pm 2.2}$ |
| LLM Debate (Du et al., 2023) | $\mathbf{39.0 \pm 3.4}$ | $43.6 \pm 3.4$ | $17.4 \pm 2.6$ | $76.0 \pm 3.0$ | $88.9 \pm 2.2$ |
| Step-back Abstraction (Zheng et al., 2023) | $31.1 \pm 3.2$ | $31.5 \pm 3.3$ | $12.2 \pm 2.3$ | $76.1 \pm 3.0$ | $87.8 \pm 2.3$ |
| Quality-Diversity (Lu et al., 2024c) | $23.8 \pm 3.0$ | $28.0 \pm 3.1$ | $14.1 \pm 2.4$ | $69.8 \pm 3.2$ | $80.1 \pm 2.8$ |
| Role Assignment (Xu et al., 2023) | $30.1 \pm 3.2$ | $37.0 \pm 3.4$ | $\mathbf{18.0 \pm 2.7}$ | $73.0 \pm 3.0$ | $83.1 \pm 2.6$ |
| **Top Agents Searched on MGSM (Math)** | | **Transferred within Math Domains** | | | |
| Dynamic Role-Playing Architecture | $\mathbf{53.4 \pm 3.5}$ | $69.5 \pm 3.2$ | $\mathbf{31.2 \pm 3.2}$ | $81.5 \pm 2.6$ | $\mathbf{91.8 \pm 1.8}$ |
| Structured Multimodal Feedback Loop | $50.2 \pm 3.5$ | $64.5 \pm 3.4$ | $30.1 \pm 3.2$ | $\mathbf{82.6 \pm 2.6}$ | $89.9 \pm 2.1$ |
| Interactive Multimodal Feedback Loop | $47.4 \pm 3.5$ | $64.9 \pm 3.3$ | $27.6 \pm 3.2$ | $80.6 \pm 2.8$ | $89.8 \pm 2.1$ |

Table 5: **Performance across multiple domains when transferring top agents from the Math (MGSM) domain to non-math domains.** Agents discovered by Meta Agent Search in the math domain can outperform or match the performance of baselines after being transferred to domains beyond math. We report the test accuracy and the 95% bootstrap confidence interval.

| Agent Name | Accuracy (%) | F1 Score | Accuracy (%) | |
| --- | --- | --- | --- | --- |
| | Math | Reading Comprehension | Multi-task | Science |
| **Manually Designed Agents** | | | | |
| Chain-of-Thought (Wei et al., 2022) | $28.0 \pm 3.1$ | $64.2 \pm 0.9$ | $65.4 \pm 3.3$ | $29.2 \pm 3.1$ |
| COT-SC (Wang et al., 2023b) | $28.2 \pm 3.1$ | $64.4 \pm 0.8$ | $\mathbf{65.9 \pm 3.2}$ | $30.5 \pm 3.2$ |
| Self-Refine (Madaan et al., 2024) | $27.5 \pm 3.1$ | $59.2 \pm 0.9$ | $63.5 \pm 3.4$ | $\mathbf{31.6 \pm 3.2}$ |
| LLM Debate (Du et al., 2023) | $\mathbf{39.0 \pm 3.4}$ | $60.6 \pm 0.9$ | $65.6 \pm 3.3$ | $31.4 \pm 3.2$ |
| Step-back Abstraction (Zheng et al., 2023) | $31.1 \pm 3.2$ | $60.4 \pm 1.0$ | $65.1 \pm 3.3$ | $26.9 \pm 3.0$ |
| Quality-Diversity (Lu et al., 2024c) | $23.8 \pm 3.0$ | $61.8 \pm 0.9$ | $65.1 \pm 3.1$ | $30.2 \pm 3.1$ |
| Role Assignment (Xu et al., 2023) | $30.1 \pm 3.2$ | $\mathbf{65.8 \pm 0.9}$ | $64.5 \pm 3.3$ | $31.1 \pm 3.1$ |
| **Top Agents Searched on Math (MGSM)** | | **Transferred beyond Math Domains** | | |
| Dynamic Role-Playing Architecture | $\mathbf{53.4 \pm 3.5}$ | $70.4 \pm 0.9$ | $62.4 \pm 3.4$ | $28.6 \pm 3.1$ |
| Structured Multimodal Feedback Loop | $50.2 \pm 3.5$ | $70.4 \pm 0.9$ | $\mathbf{67.0 \pm 3.2}$ | $28.7 \pm 3.1$ |
| Interactive Multimodal Feedback Loop | $47.4 \pm 3.5$ | $\mathbf{71.9 \pm 0.8}$ | $64.8 \pm 3.3$ | $\mathbf{29.9 \pm 3.2}$ |

We transfer the discovered agent from the MGSM (Math) domain to other math domains to test whether the invented agents can generalize across different domains. Similarly, we test the top

3 agents from MGSM and transfer them to (1) four popular math domains: GSM8K (Cobbe et al., 2021), GSM-Hard (Gao et al., 2023), SVAMP (Patel et al., 2021), and ASDiv (Miao et al., 2020) and (2) three domains beyond math adopted in Section 4.2. As shown in Table 4, we observe a similar superiority in the performance of Meta Agent Search compared to baselines. More surprisingly, we observe that agents discovered in the math domain can be transferred to non-math domains (Table 5). While the performance of agents originally searched in the math domain does not fully match that of agents specifically designed for the target domains, they still outperform (in Reading Comprehension and Multi-task) or match (in Science) the state-of-the-art hand-designed agent baselines. These results illustrate that Meta Agent Search can discover generalizable design patterns and agentic systems.

## C  PROMPTS

We use the following prompts for the meta agent in Meta Agent Search. Variables in the prompts that vary depending on domains and iterations are highlighted.

We use the following system prompt for every query in the meta agent.

> **System prompt for the meta agent.**
>
> You are a helpful assistant. Make sure to return in a WELL-FORMED JSON object.

We use the following prompt for the meta agent to design the new agent based on the archive of previously discovered agents.

> **Main prompt for the meta agent.**
>
> You are an expert machine learning researcher testing various agentic systems. Your objective is to design building blocks such as prompts and workflows within these systems to solve complex tasks. Your aim is to design an optimal agent performing well on [Brief Description of the Domain].
>
> [Framework Code]
>
> [Output Instructions and Examples]
>
> [Discovered Agent Archive] (initialized with baselines, updated at every iteration)
>
> # Your task
> You are deeply familiar with prompting techniques and the agent works from the literature. Your goal is to maximize the specified performance metrics by proposing interestingly new agents.
> Observe the discovered agents carefully and think about what insights, lessons, or stepping stones can be learned from them.
> Be creative when thinking about the next interesting agent to try. You are encouraged to draw inspiration from related agent papers or academic papers from other research areas.
> Use the knowledge from the archive and inspiration from academic literature to propose the next interesting agentic system design.
> THINK OUTSIDE THE BOX.

The domain descriptions are available in Appendices E and F and the framework code is available in Appendix D. We use the following prompt to instruct and format the output of the meta agent. Here, we collect and present some common mistakes that the meta agent may make in the prompt. We found it effective in improving the quality of the generated code. These formatting prompts are inspired by Lu et al. (2024a).

> **Output Instruction and Example.**
>
> # Output Instruction and Example:
> The first key should be ("thought"), and it should capture your thought process for designing the next function. In the "thought" section, first reason about what the next interesting agent to try should be, then describe your reasoning and the overall concept behind the agent design, and finally detail

the implementation steps. The second key ("name") corresponds to the name of your next agent architecture. Finally, the last key ("code") corresponds to the exact "forward()" function in Python code that you would like to try. You must write COMPLETE CODE in "code": Your code will be part of the entire project, so please implement complete, reliable, reusable code snippets.

Here is an example of the output format for the next agent:
{"thought": "**Insights:** Your insights on what should be the next interesting agent. **Overall Idea:** your reasoning and the overall concept behind the agent design. **Implementation:** describe the implementation step by step.",
"name": "Name of your proposed agent",
"code": "def forward(self, taskInfo): # Your code here"}

## WRONG Implementation examples:
[Examples of potential mistakes the meta agent may make in implementation]

---

After the first response from the meta agent, we perform two rounds of self-reflection to make the generated agent novel and error-free (Shinn et al., 2023; Madaan et al., 2024).

---

**Prompt for self-reflection round 1.**

[Generated Agent from Previous Iteration]
Carefully review the proposed new architecture and reflect on the following points:

1. **Interestingness**: Assess whether your proposed architecture is interesting or innovative compared to existing methods in the archive. If you determine that the proposed architecture is not interesting, suggest a new architecture that addresses these shortcomings.
- Make sure to check the difference between the proposed architecture and previous attempts.
- Compare the proposal and the architectures in the archive CAREFULLY, including their actual differences in the implementation.
- Decide whether the current architecture is innovative.
- USE CRITICAL THINKING!

2. **Implementation Mistakes**: Identify any mistakes you may have made in the implementation. Review the code carefully, debug any issues you find, and provide a corrected version. REMEMBER checking "## WRONG Implementation examples" in the prompt.

3. **Improvement**: Based on the proposed architecture, suggest improvements in the detailed implementation that could increase its performance or effectiveness. In this step, focus on refining and optimizing the existing implementation without altering the overall design framework, except if you want to propose a different architecture if the current is not interesting.
- Observe carefully about whether the implementation is actually doing what it is supposed to do.
- Check if there is redundant code or unnecessary steps in the implementation. Replace them with effective implementation.
- Try to avoid the implementation being too similar to the previous agent.

And then, you need to improve or revise the implementation, or implement the new proposed architecture based on the reflection.

Your response should be organized as follows:

"reflection": Provide your thoughts on the interestingness of the architecture, identify any mistakes in the implementation, and suggest improvements.
"thought": Revise your previous proposal or propose a new architecture if necessary, using the same format as the example response.
"name": Provide a name for the revised or new architecture. (Don't put words like "new" or "improved" in the name.)
"code": Provide the corrected code or an improved implementation. Make sure you actually implement your fix and improvement in this code.

---

**Prompt for self-reflection round 2.**

Using the tips in "## WRONG Implementation examples" section, further revise the code.
Your response should be organized as follows:
Include your updated reflections in the "reflection". Repeat the previous "thought" and "name". Update the corrected version of the code in the "code" section.

---

When an error is encountered during the execution of the generated code, we conduct a reflection and re-run the code. This process is repeated up to five times if errors persist. Here is the prompt we use to self-reflect any runtime error:

---

**Prompt for self-reflection when a runtime error occurs.**

Error during evaluation:
[Runtime errors]
Carefully consider where you went wrong in your latest implementation. Using insights from previous attempts, try to debug the current code to implement the same thought. Repeat your previous thought in "thought", and put your thinking for debugging in "debug_thought".

---

## D  FRAMEWORK CODE

In this paper, we provide the meta agent with a simple framework to implement basic functions, such as querying Foundation Models (FMs) and formatting prompts. The framework consists of fewer than 100 lines of code (excluding comments). In this framework, we encapsulate every piece of information into a namedtuple Info object, making it easy to combine different types of information (e.g., FM responses, results from tool function calls, task descriptions) and facilitate communication between different modules. Additionally, in the FM module, we automatically construct the prompt by concatenating all input Info objects into a structured format, with each Info titled by its metadata (e.g., name, author). **Throughout the appendix, we renamed some variables in the code to match the terminologies used in the main text.**

Code 1: The simple framework used in Meta-Agent Search.

```python
# Named tuple for holding task information
Info = namedtuple('Info', ['name', 'author', 'content', 'iteration_idx'])

# Format instructions for FM response
FORMAT_INST = lambda request_keys: f"Reply EXACTLY with the following
    JSON format.\n{str(request_keys)}\nDO NOT MISS ANY FIELDS AND MAKE
    SURE THE JSON FORMAT IS CORRECT!\n"

# Description of the role of the FM Module
ROLE_DESC = lambda role: f"You are a {role}."

@backoff.on_exception(backoff.expo, openai.RateLimitError)
def get_json_response_from_gpt(msg, model, system_message, temperature):
    \"""
    Function to get JSON response from GPT model.

    Args:
    - msg (str): The user message.
    - model (str): The model to use.
    - system_message (str): The system message.
    - temperature (float): Sampling temperature.

    Returns:
    - dict: The JSON response.
    \"""
    ...
    return json_dict

class FM_Module:
```

```
28      \"""
29      Base class for an FM module.
30
31      Attributes:
32      - output_fields (list): Fields expected in the output.
33      - name (str): Name of the FM module.
34      - role (str): Role description for the FM module.
35      - model (str): Model to be used.
36      - temperature (float): Sampling temperature.
37      - id (str): Unique identifier for the FM module instance.
38      \"""
39
40      def __init__(self, output_fields: list, name: str, role='helpful
            assistant', model='gpt-3.5-turbo-0125', temperature=0.5) -> None:
41          ...
42
43      def generate_prompt(self, input_infos, instruction) -> str:
44          \"""
45          Generates a prompt for the FM.
46
47          Args:
48          - input_infos (list): List of input information.
49          - instruction (str): Instruction for the task.
50
51          Returns:
52          - tuple: System prompt and user prompt.
53
54          An example of generated prompt:
55          ""
56          You are a helpful assistant.
57
58          # Output Format:
59          Reply EXACTLY with the following JSON format.
60          ...
61
62          # Your Task:
63          You will given some number of paired example inputs and outputs.
                The outputs ...
64
65          ### thinking #1 by Chain-of-Thought hkFo (yourself):
66          ...
67
68          # Instruction:
69          Please think step by step and then solve the task by writing the
                code.
70          ""
71          \"""
72          ...
73          return system_prompt, prompt
74
75      def query(self, input_infos: list, instruction, iteration_idx=-1) ->
            list[Info]:
76          \"""
77          Queries the FM with provided input information and instruction.
78
79          Args:
80          - input_infos (list): List of input information.
81          - instruction (str): Instruction for the task.
82          - iteration_idx (int): Iteration index for the task.
83
84          Returns:
85          - output_infos (list[Info]): Output information.
86          \"""
87          ...
88          return output_infos
```

```
89
90      def __repr__(self):
91          return f"{self.agent_name} {self.id}"
92
93      def __call__(self, input_infos: list, instruction, iteration_idx=-1):
94          return self.query(input_infos, instruction, iteration_idx=
                iteration_idx)
95
96  class AgentSystem:
97      def forward(self, taskInfo) -> Union[Info, str]:
98          \"""
99          Placeholder method for processing task information.
100
101          Args:
102          - taskInfo (Info): Task information.
103
104          Returns:
105          - Answer (Union[Info, str]): Your FINAL Answer. Return either a
                namedtuple Info or a string for the answer.
106          \"""
107          pass
```

With the provided framework, an agent can be easily defined with a "forward" function. Here we show an example of implementing self-reflection using the framework.

Code 2: Self-Reflection implementation example

```
1   def forward(self, taskInfo):
2       # Instruction for initial reasoning
3       cot_initial_instruction = "Please think step by step and then solve
            the task."
4
5       # Instruction for reflecting on previous attempts and feedback to
            improve
6       cot_reflect_instruction = "Given previous attempts and feedback,
            carefully consider where you could go wrong in your latest
            attempt. Using insights from previous attempts, try to solve the
            task better."
7       cot_module = FM_Module(['thinking', 'answer'], 'Chain-of-Thought')
8
9       # Instruction for providing feedback and correcting the answer
10      critic_instruction = "Please review the answer above and criticize on
             where might be wrong. If you are absolutely sure it is correct,
            output 'True' in 'correct'."
11      critic_module = FM_Module(['feedback', 'correct'], 'Critic')
12
13      N_max = 5 # Maximum number of attempts
14
15      # Initial attempt
16      cot_inputs = [taskInfo]
17      thinking, answer = cot_module(cot_inputs, cot_initial_instruction, 0)
18
19      for i in range(N_max):
20          # Get feedback and correct status from the critic
21          feedback, correct = critic_module([taskInfo, thinking, answer],
                critic_instruction, i)
22          if correct.content == 'True':
23              break
24
25          # Add feedback to the inputs for the next iteration
26          cot_inputs.extend([thinking, answer, feedback])
27
28          # Reflect on previous attemps and refine the answer
29          thinking, answer = cot_module(cot_inputs, cot_reflect_instruction
                , i + 1)
```

```
30        return answer
```

# E EXPERIMENT DETAILS FOR ARC CHALLENGE

Example Input-output grid #1

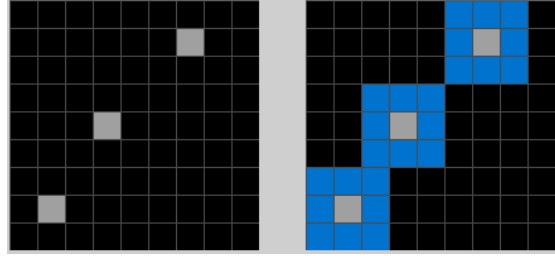

Test grid

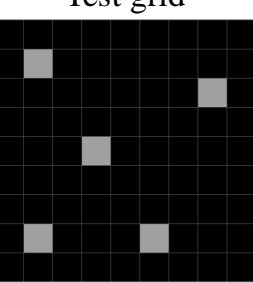

Example Input-output grid #2

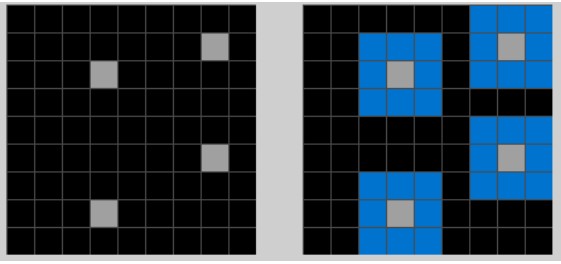

Answer

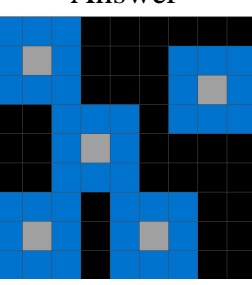

Figure 4: **An example task from the ARC challenge (Chollet, 2019).** Given the input-output grid examples, the AI system is asked to learn the transformation rules and then apply these learned rules to the test grid to predict the final answer.

An example task from the ARC challenge is shown in Figure 4. In the ARC challenge experiments (Section 4.1), we represent the grids as strings of 2-D arrays, where each color is represented by an integer. We instruct the meta agent to design agents that generate code as solutions rather than directly outputting answers. Additionally, we provide two tool functions within the framework: (1) to test whether the generated code can solve the example grids and (2) to obtain the task's answer by applying the generated code to the test grid. The accuracy rate is calculated by the Exact Match between the reference solution and the predicted answer. The meta agent uses "gpt-4o-2024-05-13" (OpenAI, 2024), while discovered agents and baselines are evaluated using "gpt-3.5-turbo-0125" (OpenAI, 2022) to reduce compute cost.

The domain description of ARC for the meta agent is shown below:

> **Description of ARC for the meta agent.**
>
> Your aim is to find an optimal agent performing well on the ARC (Abstraction and Reasoning Corpus) challenge.
> In this challenge, each task consists of three demonstration examples, and one test example. Each Example consists of an "input grid" and an "output grid". Test-takers need to use the transformation rule learned from the examples to predict the output grid for the test example.
>
> # An example task from ARC challenge:
>
> ## Task Overview:
> You will be given some number of paired example inputs and outputs grids. The outputs were produced by applying a transformation rule to the input grids. In addition to the paired example inputs and

outputs, there is also one test input without a known output.

The inputs and outputs are each "grids". A grid is a rectangular matrix of integers between 0 and 9 (inclusive). Each number corresponds to a color. 0 is black.

Your task is to determine the transformation rule from examples and find out the answer, involving determining the size of the output grid for the test and correctly filling each cell of the grid with the appropriate color or number.

The transformation only needs to be unambiguous and applicable to the example inputs and the test input. It doesn't need to work for all possible inputs. Observe the examples carefully, imagine the grid visually, and try to find the pattern.

## Examples:
### Example 0:
input = [[0,0,0,0,5,0,0,0,0], [0,0,0,0,5,0,0,0,0], [0,0,0,4,5,0,0,0,0], [0,0,0,4,5,4,4,0,0], [0,0,3,3,5,0,0,0,0], [0,0,0,3,5,0,0,0,0], [0,0,0,3,5,3,3,3,0], [0,0,0,3,5,0,0,0,0], [0,0,0,0,5,0,0,0,0], [0,0,0,0,5,0,0,0,0]]
output = [[0,0,0,0], [0,0,0,0], [0,0,0,4], [0,0,4,4], [0,0,3,3], [0,0,0,3], [0,3,3,3], [0,0,0,3], [0,0,0,0], [0,0,0,0]]

### Example 1:
input = [[0,0,0,0,5,0,0,0,0], [0,0,0,2,5,0,0,0,0], [0,0,0,2,5,2,6,0,0], [0,0,0,2,5,0,0,0,0], [0,0,0,2,5,2,2,2,0], [0,0,6,6,5,6,0,0,0], [0,0,0,2,5,0,0,0,0], [0,2,2,0,5,2,0,0,0], [0,0,0,2,5,0,0,0,0], [0,0,0,0,5,0,0,0,0]]
output = [[0,0,0,0], [0,0,0,2], [0,0,6,2], [0,0,0,2], [0,2,2,2], [0,0,6,6], [0,0,0,2], [0,2,2,2], [0,0,0,2], [0,0,0,0]]

### Example 2:
input = [[0,0,0,0,5,0,0,0,0], [0,0,0,0,5,7,0,0,0], [0,0,0,8,5,0,0,0,0], [0,0,0,8,5,0,0,0,0], [0,7,8,8,5,0,0,0,0], [0,0,0,0,5,8,8,0,0], [0,0,0,8,5,0,0,0,0], [0,0,0,8,5,0,0,0,0], [0,0,0,0,5,8,7,0,0], [0,0,0,0,5,0,0,0,0]]
output= [[0,0,0,0], [0,0,0,7], [0,0,0,8], [0,0,0,8], [0,7,8,8], [0,0,8,8], [0,0,0,8], [0,0,0,8], [0,0,7,8], [0,0,0,0]]

### Test Problem:
input = [[0,0,0,0,5,0,0,0,0], [0,0,0,1,5,0,0,0,0], [0,0,0,1,5,1,0,0,0], [0,1,1,1,5,1,1,1,6], [0,0,0,6,5,6,6,0,0], [0,0,0,0,5,1,1,1,0], [0,0,0,1,5,0,0,0,0], [0,0,0,1,5,1,6,0,0], [0,0,0,0,5,6,0,0,0], [0,0,0,0,5,0,0,0,0]]

Analyze the transformation rules based on the provided Examples and determine what the output should be for the Test Problem.

Here we present the best agent on ARC discovered by Meta Agent Search.

Code 3: The best agent on ARC discovered by Meta Agent Search

```
1  # Structured Feedback and Ensemble Agent
2  def forward(self, taskInfo):
3      # Step 1: Generate initial candidate solutions using multiple FM
              Modules
4      initial_instruction = 'Please think step by step and then solve the
              task by writing the code.'
5      num_candidates = 5  # Number of initial candidates
6      initial_module = [FM_Module(['thinking', 'code'], 'Initial Solution',
              temperature=0.8) for _ in range(num_candidates)]
7
8      initial_solutions = []
9      for i in range(num_candidates):
10         thoughts = initial_module[i]([taskInfo], initial_instruction)
11         thinking, code = thoughts[0], thoughts[1]
12         feedback, correct_examples, wrong_examples = self.
              run_examples_and_get_feedback(code)
13         if len(correct_examples) > 0:  # Only consider solutions that
              passed at least one example
14             initial_solutions.append({'thinking': thinking, 'code': code,
                  'feedback': feedback, 'correct_count': len(
                  correct_examples)})
15
16     # Step 2: Simulate human-like feedback for each candidate solution
```

```python
17      human_like_feedback_module = FM_Module(['thinking', 'feedback'], '
            Human-like Feedback', temperature=0.5)
18      human_feedback_instruction = 'Please provide human-like feedback for
            the code, focusing on common mistakes, heuristic corrections, and
             best practices.'
19
20      for sol in initial_solutions:
21          thoughts = human_like_feedback_module([taskInfo, sol['thinking'],
                sol['code']], human_feedback_instruction)
22          human_thinking, human_feedback = thoughts[0], thoughts[1]
23          sol['human_feedback'] = human_feedback
24
25      # Step 3: Assign expert advisors to evaluate and provide targeted
            feedback
26      expert_roles = ['Efficiency Expert', 'Readability Expert', '
            Simplicity Expert']
27      expert_advisors = [FM_Module(['thinking', 'feedback'], role,
            temperature=0.6) for role in expert_roles]
28      expert_instruction = 'Please evaluate the given code and provide
            targeted feedback for improvement.'
29
30      for sol in initial_solutions:
31          sol_feedback = {}
32          for advisor in expert_advisors:
33              thoughts = advisor([taskInfo, sol['thinking'], sol['code']],
                    expert_instruction)
34              thinking, feedback = thoughts[0], thoughts[1]
35              sol_feedback[advisor.role] = feedback
36          sol['expert_feedback'] = sol_feedback
37
38      # Step 4: Parse and structure the feedback to avoid redundancy and
            refine the solutions iteratively
39      max_refinement_iterations = 3
40      refinement_module = FM_Module(['thinking', 'code'], 'Refinement
            Module', temperature=0.5)
41      refined_solutions = []
42
43      for sol in initial_solutions:
44          for i in range(max_refinement_iterations):
45              combined_feedback = sol['feedback'].content + sol['
                    human_feedback'].content + ''.join([fb.content for fb in
                    sol['expert_feedback'].values()])
46              structured_feedback = ' '.join(set(combined_feedback.split())
                    )   # Avoid redundancy
47              refinement_instruction = 'Using the structured feedback,
                    refine the solution to improve its performance.'
48              thoughts = refinement_module([taskInfo, sol['thinking'], sol[
                    'code'], Info('feedback', 'Structured Feedback',
                    structured_feedback, i)], refinement_instruction, i)
49              refinement_thinking, refined_code = thoughts[0], thoughts[1]
50              feedback, correct_examples, wrong_examples = self.
                    run_examples_and_get_feedback(refined_code)
51              if len(correct_examples) > 0:
52                  sol.update({'thinking': refinement_thinking, 'code':
                        refined_code, 'feedback': feedback, 'correct_count':
                        len(correct_examples)})
53                  refined_solutions.append(sol)
54
55      # Step 5: Select the best-performing solutions and make a final
            decision using an ensemble approach
56      sorted_solutions = sorted(refined_solutions, key=lambda x: x['
            correct_count'], reverse=True)
57      top_solutions = sorted_solutions[:3]  # Select the top 3 solutions
58
```

```
59    final_decision_instruction = 'Given all the above solutions, reason
          over them carefully and provide a final answer by writing the
          code.'
60    final_decision_module = refinement_module(['thinking', 'code'], '
          Final Decision Module', temperature=0.1)
61    final_inputs = [taskInfo] + [item for solution in top_solutions for
          item in [solution['thinking'], solution['code'], solution['
          feedback']]]
62    final_thoughts = final_decision_module(final_inputs,
          final_decision_instruction)
63    final_thinking, final_code = final_thoughts[0], final_thoughts[1]
64    answer = self.get_test_output_from_code(final_code)
65    return answer
```

## F  EXPERIMENT DETAILS FOR REASONING AND PROBLEM-SOLVING DOMAINS

To reduce costs during search and evaluation, we sample subsets of data from each domain. For GPQA (Science), we use GPQA_diamond and the validation set consists of 32 questions, while the remaining 166 questions form the test set. For the other domains, the validation and test sets are sampled with 128 and 800 questions, respectively. We evaluate agents five times for GPQA and once for the other domains to maintain a consistent total number of evaluations. Each domain uses zero-shot style questions, except DROP (Reading Comprehension), which uses one-shot style questions following the practice in (OpenAI, 2023). The meta agent uses "gpt-4o-2024-05-13" (OpenAI, 2024), while discovered agents and baselines are evaluated using "gpt-3.5-turbo-0125" (OpenAI, 2022) to reduce compute cost.

We present the description of each domain we provide to the meta agent.

---

**Description of DROP (Reading Comprehension).**

Your aim is to find an optimal agent performing well on the Reading Comprehension Benchmark Requiring Discrete Reasoning Over Paragraphs (DROP), which assesses the ability to perform discrete reasoning and comprehend detailed information across multiple paragraphs.

## An example question from DROP:

You will be asked to read a passage and answer a question.

Passage:
Non-nationals make up more than half of the population of Bahrain, with immigrants making up about 55% of the overall population. Of those, the vast majority come from South and Southeast Asia: according to various media reports and government statistics dated between 2005-2009 roughly 290,000 Indians, 125,000 Bangladeshis, 45,000 Pakistanis, 45,000 Filipinos, and 8,000 Indonesians.

Question:  What two nationalities had the same number of people living in Bahrain between 2005-2009?
Answer [Not Given]: Pakistanis and Filipinos

---

**Description of GPQA (Science) for the meta agent.**

Your aim is to find an optimal agent performing well on the GPQA (Graduate-Level Google-Proof Q&A Benchmark). This benchmark consists of challenging multiple-choice questions across the domains of biology, physics, and chemistry, designed by domain experts to ensure high quality and difficulty.

## An example question from GPQA:

Two quantum states with energies E1 and E2 have a lifetime of $10^{-9}$ sec and $10^{-8}$ sec, respectively. We want to clearly distinguish these two energy levels. Which one of the following options could be their energy difference so that they be clearly resolved?

---

Answer choices:
$10^{-9}$ eV
$10^{-8}$ eV
$10^{-7}$ eV
$10^{-6}$ eV

Correct answer [Not provided]:
$10^{-7}$ eV

Explanation [Not provided]:
According to the uncertainty principle, Delta E* Delta t=hbar/2. Delta t is the lifetime and Delta E is the width of the energy level. With Delta t=$10^{-9}$ s==> Delta E1= 3.3 $10^{-7}$ ev. And Delta t=$10^{-11}$ s gives Delta E2=3.3$10^{-8}$ eV. Therefore, the energy difference between the two states must be significantly greater than $10^{-7}$ ev. So the answer is $10^{-4}$ ev.

### Description of MGSM (Math) for the meta agent.

Your aim is to find an optimal agent performing well on the Multilingual Grade School Math Benchmark (MGSM) which evaluates mathematical problem-solving abilities across various languages to ensure broad and effective multilingual performance.

## An example question from MGSM:

**Question**: この数学の問題を解いてください。

近所では、ペットのウサギの数がペットの犬と猫を合わせた数よりも12匹少ない。犬1匹あたり2匹の猫がおり、犬の数は60匹だとすると、全部で近所には何匹のペットがいますか？

**Answer (Not Given)**: 348

### Description of MMLU (Mult-task) for the meta agent.

Your aim is to find an optimal agent performing well on the MMLU (Massive Multitask Language Understanding) benchmark, a challenging evaluation that assesses a model's ability to answer questions across a wide range of subjects and difficulty levels. It includes subjects from STEM, social sciences, humanities, and more.

## An example question from MMLU:

Answer the following multiple-choice question.

The constellation ... is a bright W-shaped constellation in the northern sky.

(A) Centaurus
(B) Cygnus
(C) Cassiopeia
(D) Cepheus

## G  BASELINES

In this paper, we implement five state-of-the-art hand-designed agent baselines for experiments on ARC (Section 4.1): (1) Chain-of-Thought (COT) (Wei et al., 2022), (2) Self-Consistency with Chain-of-Thought (COT-SC)(Wang et al., 2023b), (3) Self-Refine (Madaan et al., 2024; Shinn et al., 2023), (4) LLM-Debate (Du et al., 2023), and (5) Quality-Diversity, a simplified version of Intelligent Go-Explore (Lu et al., 2024c).

In addition to these baselines, we implement two more for experiments on Reasoning and Problem-Solving domains (Section 4.2): (6) Step-back Abstraction (Zheng et al., 2023) and (7) Role Assignment (Xu et al., 2023). An example implementation of Self-Refine with our simple framework is shown in Appendix D.

In COT, we prompt the FM to think step by step before answering the question. In COT-SC, we sample $N = 5$ answers and then perform an ensemble using either majority voting or an FM query. In Self-Refine, we allow up to five refinement iterations, with an early stop if the critic deems the answer correct. In LLM-Debate, each debate module is assigned a unique role, such as Physics Expert or Chemistry Expert, and the debate lasts for two rounds. In Quality-Diversity, we conduct three iterations to collect diverse answers based on previously proposed ones. In Role Assignment, we use an FM query to first choose a role from a predefined set, and then use another FM query to answer the question by acting within the chosen role.

## H  EXAMPLE AGENTS

In this section, we present the detailed implementation of three example discovered agents by Meta Agent Search shown in Figure 1. The "Multi-Step Peer Review Agent" and "Divide and Conquer Agent" were discovered during the search in the Reading Comprehension domain (GPQA) (Rein et al., 2023), while the "Verified Multimodal Agent" was discovered during the search in the Math domain (MGSM) (Shi et al., 2023).

Code 4: Example discovered agent: Multi-Step Peer Review Agent

```
1  def forward(self, taskInfo):
2      initial_instruction = "Please think step by step and then solve the
           task."
3      critique_instruction = "Please review the answer above and provide
           feedback on where it might be wrong. If you are absolutely sure
           it is correct, output 'True' in 'correct'."
4      refine_instruction = "Given previous attempts and feedback, carefully
            consider where you could go wrong in your latest attempt. Using
           insights from previous attempts, try to solve the task better."
5      final_decision_instruction = "Given all the above thinking and
           answers, reason over them carefully and provide a final answer."
6
7      FM_modules = [FM_module(['thinking', 'answer'], 'FM Module', role=
           role) for role in ['Physics Expert', 'Chemistry Expert', 'Biology
            Expert', 'Science Generalist']]
8      critic_modules = [FM_module(['feedback', 'correct'], 'Critic', role=
           role) for role in ['Physics Critic', 'Chemistry Critic', 'Biology
            Critic', 'General Critic']]
9      final_decision_module = FM_module(['thinking', 'answer'], 'Final
           Decision', temperature=0.1)
10
11     all_thinking = [[] for _ in range(len(FM_modules))]
12     all_answer = [[] for _ in range(len(FM_modules))]
13     all_feedback = [[] for _ in range(len(FM_modules))]
14
15     for i in range(len(FM_modules)):
16         thinking, answer = FM_modules[i]([taskInfo], initial_instruction)
17         all_thinking[i].append(thinking)
18         all_answer[i].append(answer)
19
20     for i in range(len(FM_modules)):
21         for j in range(len(FM_modules)):
22             if i != j:
23                 feedback, correct = critic_modules[j]([taskInfo,
                       all_thinking[i][0], all_answer[i][0]],
                       critique_instruction)
24                 all_feedback[i].append(feedback)
25
26     for i in range(len(FM_modules)):
27         refine_inputs = [taskInfo, all_thinking[i][0], all_answer[i][0]]
               + all_feedback[i]
28         thinking, answer = FM_modules[i](refine_inputs,
               refine_instruction)
29         all_thinking[i].append(thinking)
```

```
30          all_answer[i].append(answer)
31
32      final_inputs = [taskInfo] + [all_thinking[i][1] for i in range(len(
            FM_modules))] + [all_answer[i][1] for i in range(len(FM_modules))
            ]
33      thinking, answer = final_decision_module(final_inputs,
            final_decision_instruction)
34
35      return answer
```

Code 5: Example discovered agent: Divide and Conquer Agent

```
1  def forward(self, taskInfo):
2      # Step 1: Decompose the problem into sub-problems
3      decomposition_instruction = "Please decompose the problem into
            smaller, manageable sub-problems. List each sub-problem clearly."
4      decomposition_module = FM_Module(['thinking', 'sub_problems'], '
            Decomposition Module')
5
6      # Step 2: Assign each sub-problem to a specialized expert
7      sub_problem_instruction = "Please think step by step and then solve
            the sub-problem."
8      specialized_experts = [FM_Module(['thinking', 'sub_solution'], '
            Specialized Expert', role=role) for role in ['Physics Expert', '
            Chemistry Expert', 'Biology Expert', 'General Expert']]
9
10     # Step 3: Integrate the sub-problem solutions into the final answer
11     integration_instruction = "Given the solutions to the sub-problems,
            integrate them to provide a final answer to the original problem.
            "
12     integration_module = FM_Module(['thinking', 'answer'], 'Integration
            Module', temperature=0.1)
13
14     # Decompose the problem
15     thinking, sub_problems = decomposition_module([taskInfo],
            decomposition_instruction)
16
17     # Ensure sub_problems is a string and split into individual sub-
            problems
18     sub_problems_list = sub_problems.content.split('\n') if isinstance(
            sub_problems.content, str) else []
19
20     # Solve each sub-problem
21     sub_solutions = []
22     for i, sub_problem in enumerate(sub_problems_list):
23         sub_problem_info = Info('sub_problem', decomposition_module.
                __repr__(), sub_problem, i)
24         sub_thinking, sub_solution = specialized_experts[i % len(
                specialized_experts)]([sub_problem_info],
                sub_problem_instruction)
25         sub_solutions.append(sub_solution)
26
27     # Integrate the sub-problem solutions
28     integration_inputs = [taskInfo] + sub_solutions
29     thinking, answer = integration_module(integration_inputs,
            integration_instruction)
30
31     return answer
```

Code 6: Example discovered agent: Verified Multimodal Agent

```
1  def forward(self, taskInfo):
2      # Instruction for generating visual representation of the problem
3      visual_instruction = "Please create a visual representation (e.g.,
            diagram, graph) of the given problem."
```

```
4
5      # Instruction for verifying the visual representation
6      verification_instruction = "Please verify the accuracy and relevance
           of the visual representation. Provide feedback and suggestions
           for improvement if necessary."
7
8      # Instruction for solving the problem using the verified visual aid
9      cot_instruction = "Using the provided visual representation, think
           step by step and solve the problem."
10
11     # Instantiate the visual representation module, verification module,
           and Chain-of-Thought module
12     visual_module = FM_Module(['visual'], 'Visual Representation Module')
13     verification_module = FM_Module(['feedback', 'verified_visual'], '
           Verification Module')
14     cot_module = FM_Module(['thinking', 'answer'], 'Chain-of-Thought
           Module')
15
16     # Generate the visual representation of the problem
17     visual_output = visual_module([taskInfo], visual_instruction)
18     visual_representation = visual_output[0]  # Using Info object
           directly
19
20     # Verify the visual representation
21     feedback, verified_visual = verification_module([taskInfo,
           visual_representation], verification_instruction)
22
23     # Use the verified visual representation to solve the problem
24     thinking, answer = cot_module([taskInfo, verified_visual],
           cot_instruction)
25     return answer
```

## I  PSEUDOCODE OF THE META AGENT SEARCH

In this section, we provide the pseudocode for the Meta Agent Search algorithm to clarify its implementation and workflow. The pseudocode outlines the iterative process of designing, evaluating, and refining agents using a meta agent, as described in the main text.

---

**Algorithm 1** Meta Agent Search Algorithm

---

1: **Input:** Target domain validation data, maximum iterations $N$
2: **Output:** Archive of discovered agents
3: Initialize archive $\mathcal{A}$ with baseline agents (e.g., Chain-of-Thought, Self-Refine)
4: **for** $i = 1$ to $N$ **do**
5:     **Design Step:** Meta agent generates a new agent:
6:         (a) Outputs design reasoning
7:         (b) Implements the design in code
8:         (c) Performs two self-reflection steps to ensure novelty and correctness
9:     **Evaluation Step:** Evaluate the new agent on target domain validation data:
10:         (a) If the agent produces errors during evaluation, refine the design up to 5 iterations
11:         (b) Re-run the evaluation after each refinement
12:     **Update Step:** Add the refined agent and its evaluation metrics to the archive $\mathcal{A}$
13: **end for**
14: **Return:** Final archive $\mathcal{A}$

---

## J  IMPACT OF INITIALIZATION

One of the key claims of our work is that the *code space* representation allows for better utilization of existing human efforts (Section 2), enabling a more efficient search process than starting entirely

from scratch. To further investigate the effects of initialization, we conducted experiments where the Meta Agent Search algorithm was run without any initial agent designs, contrasting with our standard approach that incorporates human-designed solutions into the search process.

The results, presented in Table 6, demonstrate that even without initial agent designs, Meta Agent Search discovers agents that outperform all hand-crafted baselines across all evaluated domains. This finding underscores the robustness of our method, as it effectively leverages the inherent structure of the code space to explore and optimize agent designs.

Interestingly, while the inclusion of good initial solutions generally leads to improved performance, the math domain exhibited a unique outcome: starting from scratch resulted in superior performance. We hypothesize that the absence of predefined design patterns in this case encouraged a broader and more diverse exploration of reasoning strategies within the limited number of iterations. Such diversity appears particularly beneficial for math tasks, which demand flexible and varied approaches to reasoning.

This observation opens up an intriguing avenue for future research: exploring how the choice and quality of initialization impact search effectiveness across different domains. For instance, it would be valuable to identify conditions under which starting without initial solutions may yield performance gains, or to design strategies that combine the advantages of both initialization and broad exploration.

Table 6: **Performance comparison of Meta Agent Search with and without initial agent designs across multiple domains.** The results show that even without initialization, Meta Agent Search outperforms hand-designed baselines in all domains. However, incorporating initial solutions generally leads to better performance, except in the math domain, where starting without initialization yields superior results.

| Agent Name | F1 Score | Accuracy (%) | | |
|---|---|---|---|---|
| | Reading Comprehension | Math | Multi-task | Science |
| **State-of-the-art Hand-designed Agents** | | | | |
| Chain-of-Thought (Wei et al., 2022) | $64.2 \pm 0.9$ | $28.0 \pm 3.1$ | $65.4 \pm 3.3$ | $29.2 \pm 3.1$ |
| COT-SC (Wang et al., 2023b) | $64.4 \pm 0.8$ | $28.2 \pm 3.1$ | $65.9 \pm 3.2$ | $30.5 \pm 3.2$ |
| Self-Refine (Madaan et al., 2024) | $59.2 \pm 0.9$ | $27.5 \pm 3.1$ | $63.5 \pm 3.4$ | $\mathbf{31.6 \pm 3.2}$ |
| LLM Debate (Du et al., 2023) | $60.6 \pm 0.9$ | $39.0 \pm 3.4$ | $65.6 \pm 3.3$ | $\mathbf{31.4 \pm 3.2}$ |
| Step-back Abstraction (Zheng et al., 2023) | $60.4 \pm 1.0$ | $31.1 \pm 3.2$ | $65.1 \pm 3.3$ | $26.9 \pm 3.0$ |
| Quality-Diversity (Lu et al., 2024c) | $61.8 \pm 0.9$ | $23.8 \pm 3.0$ | $65.1 \pm 3.3$ | $30.2 \pm 3.1$ |
| Role Assignment (Xu et al., 2023) | $65.8 \pm 0.9$ | $30.1 \pm 3.2$ | $64.5 \pm 3.3$ | $31.1 \pm 3.1$ |
| **Automated Design of Agentic Systems on Different Domains** | | | | |
| Meta Agent Search (Empty Initialization) | $73.9 \pm 0.9$ | $\mathbf{67.5 \pm 3.3}$ | $68.5 \pm 3.3$ | $\mathbf{32.7 \pm 3.2}$ |
| Meta Agent Search | $\mathbf{79.4 \pm 0.8}$ | $53.4 \pm 3.5$ | $\mathbf{69.6 \pm 3.2}$ | $\mathbf{34.6 \pm 3.2}$ |

## K  COST OF EXPERIMENTS

A single run of search and evaluation on ARC (Section 4.1) costs approximately $500 USD in OpenAI API costs, while a run within the reasoning and problem-solving domains (Section 4.2) costs about $300 USD.

The primary expense comes from querying the "gpt-3.5-turbo-0125" model during the evaluation of discovered agents. Notably, the latest GPT-4 model, "gpt-4o-mini," is less than one-third the price of "gpt-3.5-turbo-0125" and offers better performance, suggesting that we could achieve improved results with Meta Agent Search at just one-third of the cost. Additionally, as discussed in Section 6, the current naive evaluation function is both expensive and overlooks valuable information. We anticipate that future work adopting more sophisticated evaluation functions could significantly reduce the cost of ADAS algorithms.

