# OpenReview forum: "Automated Design of Agentic Systems"
_ICLR.cc/2025/Conference — ICLR 2025 Poster_

### Official Review · Reviewer_gtAT · 2024-10-19

**Soundness:** 2
**Presentation:** 2
**Contribution:** 1
**Rating:** 3
**Confidence:** 3

**Summary:**

Through this paper the authors propose to "form" a new research area they call "Automated Design of Agentic Systems", in which the high level general idea is to automate the generation of "new agents" (in this case the word agent is loosely used for a piece of code, apparently), that are continuously evolved.

**Strengths:**

- Thought-inducing subject, as the authors propose to work on "agent-generating agents".
- Contemporary work, following the trend of developing approaches to improve inference once the LLM is already trained, which makes it of interest to a good portion of the community.

**Weaknesses:**

- The biggest and greatest critique to this paper is that on the high level the authors are proposing to create a sub-community dedicated to exploring what could be seen as a primitive form of self-replication. Given that in practice the proposed method is pretty much just a code generator (easy to isolate and secure by making the code run in a Domain Specific Language), I won't reject or reduce the score of the paper due to this point, but I urge the authors to reflect on whether if push for the development of such general "Agent generation procedures" is really a good idea. Conceptually speaking, it's not a big jump to go from an agent that generates code that generates other agents, to a robot that builds other robots.

- Related to the first matter, such a research should have safety as a cornerstone of the design, with clear descriptions of how to make the agent generation safe. However, the only mention of "safety" throughout the paper is pretty much the discussion on section 6, that only talks about the "danger of creating AGI" (which in my opinion is totally eclipsed by the danger of self-replication in the case of this paper), and to which the author's only response is to push to future works to think of safety, which in my opinion is completely inappropriate. Every single building block of the algorithm should have been thought of in terms of safety and clear recommendations to avoid dangers should have been given.

- The practical description of the method is very unclear and far too high-level in the main text. I cannot understand from the description exactly what type of code is generated and what are the building blocks that are given as prior to the agent. The expanded analysis of the related works should have been moved to the appendix to make room for a better explanation of the method. Overall, it seems like the authors' approach would require a lot of compute AFTER a lot of compute has already been dedicated to train the LLM, yet, the computational cost addition to the inference was not compared to the other, clearly less-computationally expensive, approaches such as chain of thought in table 1. The method has very limited applicability if it clearly cannot be applied to hundreds of billions of parameters even if you have the supercomputers for that.

- The author statement "community has not reached a consensus on the definitions or terminologies of agents" is at the very least a weird one. Certain communities have been working on agents since the 80s and while there is always space to argue about certain gray areas of the definition, textbooks used to teach undergrad students already have a pretty decent definition for "agents", for example AI a modern approach. The authors seem to be too focused in the LLM subcommunity and disregarding past work and definitions that are relevant to the work they are trying to develop.

- Unless the authors are part of openAI I suggest against using their logo in figure 2.

- Despite being at some extent intriguing to think about some aspects of the authors' work, in my opinion the work is not very technically mature, with relatively simple and overly high level descriptions, and very limited experimentation. In my judgement this paper is more suited to an ICLR workshop.

**Questions:**

no specific question.

**Details Of Ethics Concerns:**

Although very primitive, the author's high level ideas could be seem as a sort of self-replication (read first point in weaknesses)

---

> ### Author Response · Authors · 2024-11-24
> **Response to Reviewer gtAT (Part 1/2)**
>
> Thank you for your review, and for noting our paper is thought-inducing and interesting to a good portion of the community.
>
> We sincerely believe our paper presents a significant and beneficial contribution that would enrich the ML community upon publication. We greatly appreciate your comments on safety concerns and acknowledge that our initial submission did not adequately convey our strong commitment to safety. We believe we have now addressed your concerns, particularly those related to safety considerations. Your current score recommends not publishing this work, but we hope you will keep an open mind to reconsidering in light of our significant improvements to the manuscript, especially describing everything we did to mitigate safety concerns as well as explaining why the work is valuable even with-and even because of- safety issues.
>
> > Summary: The authors propose to "form" a new research area
>
> To clarify, our argument is that we are describing a newly forming research area, which includes some prior work in this direction, and then introducing a new entrant in this space that is higher performing and more general.
>
> > Weakness 1 & 2: Concerns on safety
>
> Please see the general response.
>
> > Weakness 3 Part 1: The practical description of the method is very unclear and far too high-level in the main text.
>
> Thank you for pointing this out. We have revised the relevant sections to provide a more detailed description of the algorithm in Section 3. Additionally, we have included pseudocode in Appendix I to better illustrate the process.
>
> We have added the following text in the method section to make it clearer: “The algorithm proceeds as follows: (1) The archive is (optionally) initialized with baseline agents such as Chain-of-Thought and Self-Refine. (2) Conditioned on the archive, the meta agent designs a new agent by generating a high-level description of the new idea for an agentic system and then implementing it in code. The design then undergoes two self-reflection steps by the meta agent to ensure it is novel. (3) The generated agent is evaluated using validation data from the target domain. If errors occur during evaluation, the meta agent performs a self-reflection step to refine the design, repeating this process up to five times if necessary. (4) Finally, the agent is added to the archive along with its evaluation metrics, and the process continues with the updated archive until the maximum number of iterations is reached.”
>
> > Weakness 3 Part 2: the computational cost addition to the inference was not compared to the other, clearly less computationally expensive, approaches such as chain of thought in Table 1
>
>
> Recent advancements, such as the o1-series model (OpenAI, 2024), highlight the potential of scaling compute during inference for foundation models and agents. Our algorithm is designed to explore effective ways to leverage this scalability rather than focusing on resource-constrained application scenarios. As suggested in our future work, it would be interesting and promising to investigate this further using multi-objective optimization algorithms that balance performance and computational cost.
>
> > Weakness 3 Part 3: The method has very limited applicability if it clearly cannot be applied to hundreds of billions of parameters even if you have the supercomputers for that.
>
> We believe this method has strong potential to be applied to foundation models with hundreds of billions of parameters. First, our current approach is already feasible for larger models like GPT-4o; our current limitations (e.g., using GPT-3.5) stem from financial constraints typical of academic labs, rather than limitations of the method itself, since our method demonstrates promising sample efficiency. Second, as suggested in our future work, more efficient evaluation functions could be developed to reduce computational overhead, such as sampling test entries and examining logs in detail instead of evaluating all entries. Third, our model transfer experiments demonstrate that less expensive models can serve as effective proxies for discovering agents applicable to more costly models. Finally, with the consistent trend of exponentially increasing affordability and speed of APIs (e.g., GPT-4o-mini from July 2024 is 133x cheaper than GPT-3.5 from November 2022 while being significantly more powerful), we anticipate these constraints will continue to diminish. For these reasons, we remain confident in the scalability of this method for larger foundation models.

---

> > ### Author Response · Authors · 2024-11-24
> > **Response to Reviewer gtAT (Part 2/2)**
> >
> > > Weakness 4: Definition of agents and classic agent works from 80s
> >
> > Throughout the paper, we refer specifically to agents powered by Foundation Models, a concept that has emerged very recently and is distinct from classic agent works from the 1980s. To highlight this distinction, we use the term “agentic systems” to describe these modern agents and use “agents” as a shorthand for this concept, rather than referring to the classic agent paradigm.
> >
> > > Weakness 5: Unless the authors are part of openAI I suggest against using their logo in figure 2.
> >
> > Thank you for your advice. We replaced the logo as you suggested.
> >
> > > Weakness 6: the work is not very technically mature, with relatively simple and overly high level descriptions, and very limited experimentation
> >
> > In response to your feedback, we have expanded the methodological descriptions to provide greater technical depth and clarity. Additionally, we have broadened the experimental scope by including two new sets of experiments suggested by reviewers. Our updated results now cover extensive evaluations across nine widely used benchmarks, comparisons against eight state-of-the-art baselines, assessments using four foundation models, and analyses of the impact of initialization and transferability across both domains and models. These additions aim to address the concern about limited experimentation and enhance the technical maturity of the work.
> >
> > ---
> >
> > Have we adequately addressed your concerns, particularly around safety and our commitments to the safe development of our system? If so, we would appreciate it if you would consider increasing your score. If not, please let us know if there is anything else we can answer and/or address.

---

> ### Comment · Reviewer_gtAT · 2024-11-26
>
> Seems like from the authors' responses that I did not have any gross misunderstanding on the main reasons for my scoring. Therefore I keep the score.

---

> > ### Author Response · Authors · 2024-11-26
> > **Second Response to Reviewer gtAT**
> >
> > Dear Reviewer gtAT,
> >
> > We are surprised by your response and would like to understand the reasons behind your harsh judgment of our paper.
> >
> > We have invested significant effort in improving the paper’s technical depth, clarity, and experimental scope, and particularly described (and expanded our description of) our strong commitment to safety in the general response. We covered both our implemented safety measures (including containerization, manual checking, and signposting) in line with other works in this field which have been accepted at top-tier conferences, as well as a discussion on potential safety benefits and mitigations for the prospect of self-improving systems.
> >
> > Could you clarify precisely where the rebuttal and general response do not address your concerns, and what we can do to bridge that gap? What are the main reasons you believe the paper should be rejected?
> >
> > We hope to have a constructive discussion with you where we can work together to address any concerns you may have about our manuscript. Indeed, following discussions with Reviewer wS7u, they were happy to raise their score to strong accept (10). One other reviewer rates the work as an 8. Thus, some reviewers strongly agree with us that this work is ready for publication. Can you please reconsider with an open mind your score or let us know what we can do to earn a higher score?
> >
> > Thank you very much for reconsidering with an open mind.

---

### Official Review · Reviewer_955X · 2024-11-03

**Soundness:** 2
**Presentation:** 2
**Contribution:** 2
**Rating:** 3
**Confidence:** 2

**Summary:**

This paper proposes an approach that automatically searches building blocks and combinations of them as code generation for the Automated Design of Agentic Systems. Meta Agent Search, an iterative optimization method with foundation models generating new agents, is presented. The experiment shows that the Meta Agent Search is capable of inventing novel designs of agents. It is claimed that Meta Agent Search consistently outperforms state-of-the-art hand-designed agents in various domains. The optimized agents are shown to be transferable to different domains and foundation models.

**Strengths:**

1. The work is well-motivated.
2. The transferability results are interesting.

**Weaknesses:**

1. Some of the sentences are not written with care. For example, "Given that programming languages are Turing Complete...". The fact is that some programming languages are not Turning complete. This is awarded by the authors on line 106.
2. It should be stated in the main text that the implementation of the meta optimizers is different for different tasks.
3. As an optimization method, it is only compared with handcrafted baselines. More comparison with other optimization methods is expected.
4. I think there are missing citations for self-referential higher-order meta-learning. See Schmidhuber. Evolutionary principles in self-referential learning, or on learning how to learn: The meta-meta-... hook. Diploma thesis, Institut für Informatik, Technische Universität München, 1987 and its follow-up works.
5. I think there is a lack of explanation of how the work illustrates the potential of this direction to benefit humanity, which is claimed in the abstract.
6. The optimization starts with the baseline method as the initial solution. For a fair comparison, I think the optimization result starting without any manually optimized initial solution should be presented.

**Questions:**

1. How should I interpret the legend "E.g. LLM defines agents using code" in the search algorithm box of Figure 2?
2. What is "the framework" on line 265 referring to?
3. What is the motivation for reporting the median accuracy instead of the mean? Is this a common practice? Could you also so the mean?
4. The paper states that the method is compared with five state-of-the-art baselines, namely COT, COT-SC, Self-Refine, LLM-Debate, and Quality-Diversity. Could you elaborate in what sense they are state-of-the-art? Are they the best-performed known method for at least one of the tasks being tested? To my knowledge, the best handcrafted solution of ARC achieves at least 20% accuracy (see the reference below). Please specify the scope of your baselines.

Ferré, S. (2021). First steps of an approach to the arc challenge based on descriptive grid models and the minimum description length principle. arXiv preprint arXiv:2112.00848.

5. How exactly is the meta agent implemented? Is it a single query of an LLM with a handcrafted template presented in Appendix C?

**Details Of Ethics Concerns:**

The paper presents methods to generate code in Turing complete languages. As the search space is Turing complete, the generated code can, in principle, do almost anything, for example, attack a website. I think the method needs more studies about safety concerns.

---

> ### Author Response · Authors · 2024-11-24
> **Response to Reviewer 955X (Part 1/2)**
>
> Thank you for your review, and for noting our work is well-motivated and the transferability results are interesting.
>
> We believe we have addressed each of your concerns, including conducting two additional experiments you requested, significantly improving the manuscript as a result. Overall we would have predicted a higher score based just on your comments alone, as it did not feel to us that the level of issues you raised merited such a low score. We sincerely believe our paper presents a significant and beneficial contribution that would enrich the ML community upon publication. Your current score recommends not publishing this work, but we hope you will keep an open mind to reconsidering in light of our revisions, replies, and additional experiments.
>
> > Weakness 1: "Given that programming languages are Turing Complete..." is not written with care.
>
> Thank you for pointing this out. We have revised the sentence to ensure it is more precise. “Given that *most* programming languages are Turing Complete...” (Line 18)
>
> > Weakness 2: It should be stated in the main text that the implementation of the meta optimizers is different for different tasks.
>
> Our meta agent implementation is identical across different tasks, with the only variation being task-specific descriptive text included in the prompt. This is explained in Appendix C, but we have added an explanation of it in the main text too (Line 230).
>
> > Weakness 3: As an optimization method, it is only compared with handcrafted baselines. More comparison with other optimization methods is expected.
>
> Thank you for your insightful comment! Based on your suggestion, we added experimental results for a state-of-the-art prompt optimization baseline (OPRO, Yang et al., 2024) and added them to Table 1.
>
> | **Agent Name**              | **Reading Comprehension (F1)** | **Math (%)**  | **Multi-task (%)** | **Science (%)** |
> |-----------------------------|-------------------------------|---------------|---------------------|-----------------|
> | Prompt Optimization  (Yang et al. 2024)       | 69.1 ± 0.9                   | 30.6 ± 3.2    | 66.4 ± 3.2          | 32.9 ± 3.2      |
> | Meta Agent Search (Ours)    | **79.4 ± 0.8**               | **53.4 ± 3.5**| **69.6 ± 3.2**      | **34.6 ± 3.2**  |
>
> The results demonstrate that our proposed Meta Agent Search outperforms OPRO across all domains. This additional comparison further reinforces the discovery that defining agents in code and allowing the learning of all components provides significant advantages.
>
> > Weakness 4: I think there are missing citations for self-referential higher-order meta-learning.
>
> Thank you for your feedback. We have added the relevant citation [1] on self-referential higher-order meta-learning as per your suggestion in Section 6 Future Work.
>
> [1] Schmidhuber, Jurgen. "Evolutionary principles in self-referential learning." On learning how to learn: The meta-meta-... hook.) Diploma thesis, Institut f. Informatik, Tech. Univ. Munich 1.2 (1987): 48.
>
> > Weakness 5: how the work illustrates the potential of this direction to benefit humanity, which is claimed in the abstract.
>
> Please see the general response.
>
>
> > Weakness 6: For a fair comparison, I think the optimization result starting without any manually optimized initial solution should be presented.
>
> Since one of our claims is that the code space allows for better utilization of existing human efforts (Section 2) rather than starting from scratch, we believe that starting without initial solutions is not necessarily a fairer comparison. However, we agree that it is an interesting investigation to explore the effects of initialization on the archive.
>
> To address this, we conducted experiments running the Meta Agent Search without any initial agent designs. The results show that the discovered agents still outperform hand-crafted baselines across all domains. Interestingly, while good initial solutions generally improve performance, starting from scratch yielded superior results in the math domain. We hypothesize this is because the absence of initial design patterns may result in a broader exploration of different reasoning strategies within limited iterations, which is particularly beneficial in math tasks requiring diverse reasoning approaches. This opens up interesting future work on how initialization impacts search effectiveness across different domains. We have added the results and discussion in Appendix K.
>
> | **Agent Name**                     | **Reading Comprehension (F1)** | **Math (%)**  | **Multi-task (%)** | **Science (%)** |
> |------------------------------------|-------------------------------|---------------|---------------------|-----------------|
> | Meta Agent Search (Empty Initialization) | 73.9 ± 0.9                   | **67.5 ± 3.3**| 68.5 ± 3.3          | 32.7 ± 3.2      |
> | Meta Agent Search          | **79.4 ± 0.8**               | 53.4 ± 3.5 | **69.6 ± 3.2**      | **34.6 ± 3.2**  |

---

> > ### Author Response · Authors · 2024-11-24
> > **Response to Reviewer 955X (Part 2/2)**
> >
> > > Question 1: How should I interpret the legend "E.g. LLM defines agents using code" in the search algorithm box of Figure 2?
> >
> > This illustrates an example of how the search algorithm explores the search space, i.e. designs new agents. Specifically, it shows that an LLM can be used to design new agents directly in code. More specifically, the LLM looks at all past agents created and is instructed to try to create a new agent (described in code) that is interestingly different from prior agents and high-performing.
> >
> > > Question 2: What is "the framework" on line 265 referring to?
> >
> > It refers to the simple framework we provide to the meta agent, which includes basic functions such as output formatting and API query handling, as described in Section 3. We have updated the text in Section 4.1 to make it clearer.
> >
> > > Question 3: What is the motivation for reporting the median accuracy instead of the mean? Is this a common practice?
> >
> > Typically, using the median provides a more robust and reliable measure of central tendency when the data distribution is unknown or potentially non-normal. (Huber, P. J. (1981). Robust Statistics. Wiley.) It is considered a best practice as the default choice because means are more affected by outliers than medians. This is a standard choice throughout all scientific disciplines.
> >
> > > Question 4: Could you elaborate in what sense the chosen baselines are state-of-the-art?
> >
> > Thank you for pointing out that the paper was not clear enough on this. By state-of-the-art, we mean that the baseline agent designs represent important design patterns or are widely adopted across various applications. We do not claim they are the absolute SOTA agents for specific domains, such as the ARC challenge. We have updated the paper in Section 4.1 Baselines to clarify this point.
> >
> > > Question 5: How exactly is the meta agent implemented? Is it a single query of an LLM with a handcrafted template presented in Appendix C?
> >
> > The meta agent is implemented as a hand-crafted self-reflection agent. It begins with an initial query followed by two reflection steps, and up to five additional reflection steps in the case of errors, as detailed in Appendix C.
> >
> > We have added the following text in the method section to make it clearer: “The algorithm proceeds as follows: (1) The archive is (optionally) initialized with baseline agents such as Chain-of-Thought and Self-Refine. (2) Conditioned on the archive, the meta agent designs a new agent by generating a high-level description of the new idea for an agentic system and then implementing it in code. The design then undergoes two self-reflection steps by the meta agent to ensure it is novel. (3) The generated agent is evaluated using validation data from the target domain. If errors occur during evaluation, the meta agent performs a self-reflection step to refine the design, repeating this process up to five times if necessary. (4) Finally, the agent is added to the archive along with its evaluation metrics, and the process continues with the updated archive until the maximum number of iterations is reached.”
> >
> > > Ethics Concerns on safety concerns about code search space
> >
> > Please see the general response.
> >
> > ---
> >
> > Have we adequately addressed your concerns? If so, we would appreciate you considering increasing your score. If not, please let us know if there is anything else we can answer and/or address.

---

> > > ### Author Response · Authors · 2024-11-26
> > > **Follow-up on Reviewer 955X**
> > >
> > > Dear Reviewer 955X,
> > >
> > > Thank you for your detailed review and feedback. As we approach the end of the discussion period, we notice that you have not yet responded to our detailed answers to your concerns. Your score differs substantially from other reviewers who rated the paper more favorably (an accept, 8, and a strong accept, 10). Since ICLR typically requires consensus among reviewers for acceptance, your current score could prevent the paper from being published despite the positive feedback from two other reviewers.
> > >
> > > - We have worked diligently to address each of your concerns by:
> > > - Fixing writing issues (e.g., clarifying statements on Turing completeness).
> > > - Adding experiments comparing to state-of-the-art optimization methods (OPRO).
> > > - Conducting experiments without initial solutions, showing strong results.
> > > - Adding the suggested citation on self-referential meta-learning.
> > > - Clarifying our baselines and evaluation choices (median vs. mean).
> > > - Expanding details on the meta agent’s implementation and process.
> > > - Addressing ethical concerns with a safety discussion.
> > >
> > > Given these improvements and our detailed responses, would you be willing to review our rebuttal and reconsider your rating? If there are remaining concerns that would need to be addressed to merit a higher score, we would greatly appreciate your feedback before the rebuttal period ends.

---

> ### Comment · Reviewer_955X · 2024-12-01
>
> Thanks for your response. I'm satisfied with all of them except weakness number 3. For weakness number 3, your method is compared with OPRO, which is a prompt optimization method but not an agent optimization one. I think a fair compression would be with an agent optimization method with the same initial agent and search space. Given that the motivation of the paper is to effectively find better LLM agents, I believe that showing your method's performance compared to agent optimization methods such as [1, 2, 3] is essential. Therefore, I will keep my score.
>
>
> [1] Zhuge, M., Wang, W., Kirsch, L., Faccio, F., Khizbullin, D., & Schmidhuber, J. (2024). Language agents as optimizable graphs. arXiv preprint arXiv:2402.16823.
>
> [2] Cheng, C. A., Nie, A., & Swaminathan, A. (2024). Trace is the next autodiff: Generative optimization with rich feedback, execution traces, and llms. arXiv preprint arXiv:2406.16218.
>
> [3] Yuksekgonul, M., Bianchi, F., Boen, J., Liu, S., Huang, Z., Guestrin, C., & Zou, J. (2024). TextGrad: Automatic" Differentiation" via Text. arXiv preprint arXiv:2406.07496.

---

> ### Author Response · Authors · 2024-12-03
> **Second Response to Reviewer 955X**
>
> Dear Reviewer 955X,
>
> Thank you for your feedback and for agreeing that we have addressed all your concerns except for weakness 3 in your review. Is there any chance you might reconsider raising your score to reflect that we have addressed all concerns except one, since that indicates the paper is now better according to your criteria than when you first selected that score?
>
> Regarding the remaining issue, we received your request for additional experiments one day before the end of the discussion period. Had you asked at the beginning of the discussion period, we would have made every effort to incorporate these experiments. Unfortunately, we are unable to conduct new experiments at this stage due to time constraints, so we would like to explain our views regarding the issue you raised.
>
> We understand that you suggest comparing our method with agent optimization methods that use the same initial agent and search space. However, our work is the first to explore the optimization of the entire agentic system in code, which is the least constrained search space. As such, there are currently no prior works that optimize agents in code with the same initial agent and search space as ours. This makes it challenging to perform a direct comparison under identical conditions. Moreover, our main contributions lie not only in the optimization method but also in defining a new search space and problem formulation for agent optimization. We believe that controlling for “the same initial agent and search space” is not thus applicable in our context, or at least that it would risk the reader getting distracted with comparing our innovation (a much broader, general method) with more constrained search in a constrained search space. That would thus risk people missing the point of our paper, which is demonstrating that searching within the most general, powerful search space is possible.  The fact that this is true is one of the reasons we think this work is valuable: it opens up a new research frontier, rather than being a new/better way to do something others have tried before.
>
> Regarding the comparison with prompt optimization methods, we chose OPRO as a baseline because prior works primarily focus on prompt optimization for agents. Even in the references you mentioned, [1] includes prompt optimization as an independent step, and [2] and [3] also primarily optimize prompts in agents as one of the applications of their general optimization frameworks.
>
> Given the above arguments, we believe that comparing our method with OPRO is representative and provides valuable insights into how our algorithm in the proposed code search space performs relative to established prompt optimization techniques for agents. We acknowledge that incorporating additional baselines, such as [1-3], could strengthen our results, and we will make an effort to include them in the camera-ready version of the paper. However, due to time constraints, we hope our explanations suffice to address your concerns.
>
> Overall, we believe that the issues you raised do not warrant such a low score, especially given (1) that we have fixed many of the issues you asked us to since you selected that score, (2) we improved the paper substantially based on the other reviewers’ comments as well, and (3) the positive evaluations from the other reviewers (an accept with a score of 8 and a strong accept with a score of 10). We worry that the “anchoring effect” (to your original score) may be at play, and wonder if you might be open to reconsidering your score.
>
> Thank you very much for your time and for at least considering reconsidering your score a final time.
>
> The Authors

---

### Official Review · Reviewer_MJM9 · 2024-11-04

**Soundness:** 3
**Presentation:** 3
**Contribution:** 4
**Rating:** 8
**Confidence:** 3

**Summary:**

The paper introduces a new research area called Automated Design of Agentic Systems (ADAS), aiming to automate the creation of powerful agentic systems by inventing novel building blocks and combining them in innovative ways. The authors propose an approach where agents are defined in code, allowing a meta-agent to program new agents iteratively. They present an algorithm called Meta Agent Search, where a meta-agent utilizes Foundation Models (FMs) to generate, evaluate, and improve agent designs based on an ever-growing archive of previous discoveries. Through extensive experiments across multiple domains—including logic puzzles, reading comprehension, mathematics, and science—they demonstrate that agents discovered by Meta Agent Search outperform state-of-the-art hand-designed agents. Moreover, these agents show strong generalization capabilities when transferred across different domains and models.

**Strengths:**

1. The introduction of ADAS as a research area is novel, and framing the agent design problem as code generation by a meta-agent is a creative approach.
2. The Meta Agent Search algorithm is well-conceived, and the experimental results are robust, showing consistent improvements over existing methods.
3. The paper clearly articulates the motivation, methodology, and implications of the work, making it accessible to readers.
4. Automating agent design has the potential to accelerate AI development significantly, and the demonstrated generalization capabilities suggest wide applicability.

**Weaknesses:**

1. While the paper mentions safety concerns, a more in-depth analysis of potential risks and mitigation strategies associated with ADAS would be beneficial.
2. A deeper discussion on why certain baselines were chosen and how they were implemented would strengthen the evaluation.

**Questions:**

1. How does Meta Agent Search address potential biases or unsafe behaviors that might emerge from automatically generated agents? Are there mechanisms in place to detect and mitigate these issues?
2. How does the approach scale with more complex tasks or larger domains? Are there computational limitations that need to be considered?
3. Could the authors elaborate on how higher-order ADAS (e.g., improving the meta-agent itself) might be achieved and what challenges are anticipated in that direction?

---

> ### Author Response · Authors · 2024-11-24
> **Response to Reviewer MJM9**
>
> Thank you for your comprehensive review and for recognizing the strengths of our work, including the novelty of introducing ADAS as a research area, the robust experimental results, the clarity in presenting the motivation, methodology, and implications, and the potential of automating agent design to significantly accelerate AI development and broaden applicability. We appreciate your feedback and are pleased to note your positive assessment. We address your concerns and suggestions below.
>
> > Weakness 1 & Question 1: While the paper mentions safety concerns, a more in-depth analysis of potential risks and mitigation strategies associated with ADAS would be beneficial. How does Meta Agent Search address potential biases or unsafe behaviors that might emerge from automatically generated agents? Are there mechanisms in place to detect and mitigate these issues?
>
> Please see the general response.
>
> > Weakness 2: A deeper discussion on why certain baselines were chosen and how they were implemented would strengthen the evaluation.
>
> Thank you for your comment. We have added more discussions about the baselines in Section 4.1 Baselines. The chosen baselines represent agent designs widely adopted in the agent literature. For fair comparisons, we implemented them using the same framework provided for the meta agent. Example code for the baselines and related framework is included in Appendix D. All code, including the baseline implementations, will be open-sourced.
>
> Also, we have incorporated a state-of-the-art prompt optimization method (OPRO, Yang et al., 2024) as an additional baseline (Table 1). The results demonstrate that our proposed Meta Agent Search outperforms OPRO across all domains, further reinforcing our argument that defining agents in code and enabling the learning of all components provides significant advantages.
>
> > Question 2: How does the approach scale with more complex tasks or larger domains? Are there computational limitations that need to be considered?
>
> We expect the proposed Meta Agent Search to scale effectively to more complex tasks and larger domains due to its flexibility, which allows for the integration of additional tools and functionalities as needed. While computational limitations may arise in larger domains, these challenges can be addressed through optimizations such as efficient evaluation functions to balance performance and cost. Overall, the use of a flexible code search space makes Meta Agent Search a promising approach for scaling to a wide range of applications. Moreover, as foundation models and computational resources continue to exponentially decrease in cost (e.g., GPT-4o-mini from July 2024 is 133x cheaper than GPT-3.5 from November 2022 while being significantly more powerful), ADAS will become increasingly feasible for application across a broader range of complex domains.
>
> > Question 3: Could the authors elaborate on how higher-order ADAS (e.g., improving the meta-agent itself) might be achieved and what challenges are anticipated in that direction?
>
>
> That is a great question! One potential approach is to introduce a “meta-meta agent” and run a “Meta-meta Agent Search” to improve the meta agent itself. However, this would pose significant computational challenges, as each evaluation of a meta agent design would require a full run of the Meta Agent Search algorithm. An alternative is to enable online adaptation of the meta agent during the search process. While this approach may be more feasible from a cost perspective, it introduces the challenge of assessing the quality of the evolving meta agent. We also found [1,2] provides relevant insights into this intriguing topic.
>
> ---
>
> Thank you for recognizing the contribution of our work. Have we answered all your questions and concerns? Please do not hesitate to bring up any additional questions.
>
> [1] Lu, Chris, Sebastian Towers, and Jakob Foerster. "Arbitrary Order Meta-Learning with Simple Population-Based Evolution." ALIFE 2023: Ghost in the Machine: Proceedings of the 2023 Artificial Life Conference. MIT Press, 2023.
>
> [2] Lange, Robert Tjarko, et al. "Discovering Evolution Strategies via Meta-Black-Box Optimization." The Eleventh International Conference on Learning Representations.

---

> > ### Comment · Reviewer_MJM9 · 2024-12-02
> >
> > Thank you for your detailed and thoughtful responses to my comments. Your clarifications have addressed all of my concerns, and I now have a much clearer understanding of the points in question.

---

### Official Review · Reviewer_wS7u · 2024-11-04

**Soundness:** 3
**Presentation:** 3
**Contribution:** 2
**Rating:** 10
**Confidence:** 4

**Summary:**

In the style of neural architecture search, the authors argue that the design of agentic systems can be automated and provide a system that is a step toward solving that problem. They show that their system can achieve strong performance on several benchmarks compared to hand-designed systems.

**Strengths:**

- The work is decently well presented and does read like a paper one would expect to see at ICLR.
- The topic is relevant, the paper is timely, and the results are somewhat strong.
- I can see this naturally integrating with work trying to design better evaluators (LLM-as-a-Judge, Agent-as-a-Judge, etc.).

**Weaknesses:**

- My most serious concern lies with the novelty of this work. There are papers like Voyager out there which incrementally build up sets of tools for solving a specific task. Is this actually fundamentally different from such code-generating systems? Or just a new paint job?
- The authors keep citing the Turing Complete nature of their proposal. Most things in AI are, in one way or another, Turing Complete, so I wonder if that is really a big deal here. Like everything else here, the Turing Completeness of their system is wholly dependent on the components of the actual system (here, whether the LLMs actually act as Turing Complete systems in practice).
- From the main text, it is quite unclear how their proposed system works exactly. Is it just a trivial loop writing a basic program?
- I think you want to cite "The Bitter Lesson" alongside Clune in Line 50. It's more disconnected from the other works this one cites, so it is a bit less biased. While a blog post, it predates the cited work from Clune here and was quite influential. I'm sure there's earlier stuff that's good to cite here, too (this is one of the core motivations of this work, so it's worth fleshing out).
- I like the presence of the research question on lines 88-89. But given that it's so important to the text, maybe polish it a bit by refining it (maybe just drop everything from "rather" onwards).
- When bolding results, you need to bold everything that has overlapping error bars. Statistically, the results probably wouldn't pass a t-test and that's normal in the field, but we still can't get away with totally ignore statistics.

**Questions:**

Please respond to the above when possible as well.

- What exactly is the flowchart for the system here? It would be good to see something like that covering all the details in Section 3 (something more detailed than Figure 1).
- Do LLMs act as Turing Complete systems?
- What is and isn't hand-engineered here? Maybe use Figure 3(b) to explain.
- The system seems to build itself up bit by bit. This isn't exactly a nice optimization space, though, so how do you counteract local minima? Neural architecture search has quite a bit of focus on this (even the original NEAT is mainly focused on this).
- Can you clarify why you are using grids to sample in line 267?
- Can you compare this with Voyager? I'd like to see something differentiating it from generic code-generation tasks here.
- It's maybe a bit of an unfair comparison but did you compare against a vanilla GPT-o1 (I don't view this is required here)?

If the other reviewers bring up no serious issues, I can see myself changing my score if you can address most of the above.

---

> ### Author Response · Authors · 2024-11-24
> **Response to Reviewer wS7u (Part 1/3)**
>
> Thank you for your insightful and detailed review. We appreciate your recognition of our work’s strengths, including its strong results, timely and relevant topic, clear presentation, and potential integration with other significant research directions such as designing better evaluators (e.g., LLM-as-a-Judge, Agent-as-a-Judge). Your insights are encouraging and valuable to us.
>
> We believe we have addressed each of your concerns, significantly improving the manuscript as a result. We feel this paper makes an important contribution and the ML community will benefit from it if published. We appreciate your open mind to changing your score, and we hope you can improve the score given our answers to your questions and our improvements.
>
> > Weakness 1 & Question 6: There are papers like Voyager out there which incrementally build up sets of tools for solving a specific task. Is this actually fundamentally different from such code-generating systems? Can you compare this with Voyager?
>
> Both our work and Voyager involve code generation, but they differ fundamentally in focus and scope. Voyager generates code as “skills” to control a game API, enabling atomic actions without involving foundation models. In contrast, our approach generates code for agentic systems, which include complex workflows involving multiple calls to foundation models and tools. Voyager is more akin to works that generate tools within agentic systems but focuses primarily on games rather than general agentic tasks. Our framework, by contrast, is designed to search for any possible components in agentic systems (which could include Voyager-style code as tools), supporting a wide range of general tasks. Due to the differing domains and objectives, a direct comparison is impossible (e.g. a code policy for Minecraft could not answer free-form QA), but we will expand on this distinction in the related work section.
>
>
> > Weakness 2 & Question 2: Most things in AI are, in one way or another, Turing Complete, so I wonder if that is really a big deal here … whether the LLMs actually act as Turing Complete systems in practice.
>
> Turing completeness in our work specifically relates to the code search space. Unlike prior attempts to automatically discover LLM agents, which often restrict their search space (e.g., focusing solely on prompt optimization), our method defines agents in code. Because the programming language we use (Python) is Turing complete, our search space theoretically allows the discovery of all possible components of agentic systems, such as advanced tool use and RAG in open-sourced codebases like LangChain, and even beyond. As such, our system has more power versus other attempts to automatically create LLM agents. It is true that there are many systems like artificial life systems or even neural networks that are also Turing Complete, and you are right to raise the issue that having this property is not sufficient, for one needs to be able to effectively search this vast search space to produce useful artifacts. But we do show just that. Because it harnesses the powerful intelligence in Foundation Models, Meta-Agent Search marries a Turing Complete search space with the ability to search it well to discover effective solutions.
>
> > Weakness 3 & Question 1: From the main text, it is quite unclear how their proposed system works exactly. Is it just a trivial loop writing a basic program?
>
> Thank you for your comment. To improve clarity, we have added a more thorough description of the algorithm in Section 3 and pseudocode for our Meta Agent Search algorithm in Appendix I to better illustrate how the system works.
>
> We have added the following description: “The algorithm proceeds as follows: (1) The archive is (optionally) initialized with baseline agents such as Chain-of-Thought and Self-Refine. (2) Conditioned on the archive, the meta agent designs a new agent by generating a high-level description of the new idea for an agentic system and then implementing it in code. The design then undergoes two self-reflection steps by the meta agent to ensure it is novel. (3) The generated agent is evaluated using validation data from the target domain. If errors occur during evaluation, the meta agent performs a self-reflection step to refine the design, repeating this process up to five times if necessary. (4) Finally, the agent is added to the archive along with its evaluation metrics, and the process continues with the updated archive until the maximum number of iterations is reached.”

---

> > ### Author Response · Authors · 2024-11-24
> > **Response to Reviewer wS7u (Part 2/3)**
> >
> > > Weakness 4: I think you want to cite "The Bitter Lesson" alongside Clune in Line 50.
> >
> > Thank you for your comment. We have added the citation and discussed it in Line 50.
> >
> > > Weakness 5: Refines the research question on lines 88-89 (maybe just drop everything from "rather" onwards).
> >
> > Thank you. We have refined the sentence according to your suggestion in Line 88 by dropping everything from "rather" onwards.
> >
> > > Weakness 6: Bolding results and statistical test
> >
> > Thank you, and we agree. In all tables, we have revised the use of bolding to be more aligned with statistical conventions and added a description of how we bold results.
> >
> > We have added the following description: “Here, and in all tables below, we bold the entry with the highest performance for each domain, as well as all entries whose median falls within the 95% confidence interval of the highest-performing treatment.”
> >
> > > Question 3: What is and isn't hand-engineered here? Maybe use Figure 3(b) to explain.
> >
> > In our learned agent architecture, everything from prompts to workflows is automatically designed by the meta agent. Specifically, in Figure 3(b), all components—including modules, parameters such as *5* CoTs, and connections between modules—were defined by the meta agent in the code (i.e. everything was learned by the system, nothing was hand-engineered) (detailed code is available in Appendix E). In the framework we provide to the meta agent (Appendix D), we only include a basic setup to handle API queries for Foundation Models and format outputs.
> >
> > > Question 4: The system seems to build itself up bit by bit. This isn't exactly a nice optimization space, though, so how do you counteract local minima?
> >
> >
> > To address local minima, we adopt strategies from the open-endedness and quality-diversity community to encourage exploration during the search. Specifically, we guide the meta agent to discover interesting (e.g., novel or worthwhile) agents, following approaches that leverage human notions of interestingness (Zhang et al., 2024). By collecting many such interesting stepping stones (even if they are not high-performing), we eventually can discover things that are high-performing. ADAS thus has ideas within it (quality-diversity, open-endedness, goal switching, stepping stone accumulation, etc.) that have been built over a decade and have been shown to create successful exploration in even the hardest exploration environments (those that are highly sparse or deceptive) [1-6].  These ideas help mitigate challenges associated with local minima. Additionally, the “mutation” (perturbation) operator is itself a language model, so it can make intelligent leaps through the space of solutions (e.g. from one good agent to another), rather than making random changes to solutions (as happens in RL or evolutionary algorithms). That idea has been shown to work well in recent work combining ideas from open-endedness with foundation models [5-9].
> >
> >
> > > Question 5: Can you clarify why you are using grids to sample in line 267?
> >
> > Given the significant difficulty of the ARC challenge, we sampled easier questions from the dataset by selecting those with smaller input-output grid dimensions. This approach naturally simplifies the challenge and provides clearer learning signals for the agents to improve their performance.
> >
> > > Question 7: It's maybe a bit of an unfair comparison but did you compare against a vanilla GPT-o1 (I don't view this is required here)?
> >
> > Comparing against a vanilla GPT-o1 is somewhat orthogonal to our work, as ADAS focuses on improving agentic systems regardless of the foundation models used within them. In fact, ADAS could potentially be applied to enhance GPT-o1’s performance. However, due to the potential computational expense and current rate limiting, this lies outside the scope of our current study and remains an interesting direction for future work.

---

> > > ### Author Response · Authors · 2024-11-24
> > > **Response to Reviewer wS7u (Part 3/3)**
> > >
> > > > If the other reviewers bring up no serious issues, I can see myself changing my score if you can address most of the above.
> > >
> > > We appreciate your open mind to changing your score. Have we addressed all of your concerns? If not, please let us know and we will endeavor to do so. If so, we would greatly appreciate it would be willing to update your score.
> > >
> > > [1] Ecoffet, Adrien, et al. "First return, then explore." Nature 590.7847 (2021): 580-586.
> > >
> > > [2] Wang, Rui, et al. "Poet: open-ended coevolution of environments and their optimized solutions." Proceedings of the Genetic and Evolutionary Computation Conference. 2019.
> > >
> > > [3] Nguyen, Anh Mai, Jason Yosinski, and Jeff Clune. "Innovation engines: Automated creativity and improved stochastic optimization via deep learning." Proceedings of the 2015 annual conference on genetic and evolutionary computation. 2015.
> > >
> > > [4] Stanley, Kenneth O., and Joel Lehman. Why greatness cannot be planned: The myth of the objective. Springer, 2015.
> > >
> > > [5] Zhang, Jenny, et al. "OMNI: Open-endedness via Models of human Notions of Interestingness." The Twelfth International Conference on Learning Representations. 2024
> > >
> > > [6] Faldor, Maxence, et al. "OMNI-EPIC: Open-endedness via Models of human Notions of Interestingness with Environments Programmed in Code." arXiv preprint arXiv:2405.15568 (2024).
> > >
> > > [7] Lehman, Joel, et al. “Evolution through large models.” Handbook of Evolutionary Machine Learning
> > >
> > > [8] Lu, Cong, Shengran Hu, and Jeff Clune. "Intelligent Go-Explore: Standing on the Shoulders of Giant Foundation Models." arXiv preprint arXiv:2405.15143 (2024).
> > >
> > > [9] Wang, Guanzhi, et al. "Voyager: An Open-Ended Embodied Agent with Large Language Models." Transactions on Machine Learning Research.

---

> > > > ### Comment · Reviewer_wS7u · 2024-11-26
> > > > **Response to Rebuttal**
> > > >
> > > > > We appreciate your recognition of our work’s strengths, including its strong results, timely and relevant topic, clear presentation, and potential integration with other significant research directions such as designing better evaluators (e.g., LLM-as-a-Judge, Agent-as-a-Judge).
> > > >
> > > > Humility is a virtue.
> > > >
> > > > > Both our work and Voyager involve code generation, but they differ fundamentally in focus and scope. Voyager generates code as “skills” to control a game API, enabling atomic actions without involving foundation models. In contrast, our approach generates code for agentic systems, which include complex workflows involving multiple calls to foundation models and tools. Voyager is more akin to works that generate tools within agentic systems but focuses primarily on games rather than general agentic tasks. Our framework, by contrast, is designed to search for any possible components in agentic systems (which could include Voyager-style code as tools), supporting a wide range of general tasks. Due to the differing domains and objectives, a direct comparison is impossible (e.g. a code policy for Minecraft could not answer free-form QA), but we will expand on this distinction in the related work section.
> > > >
> > > > I'm still concerned with novelty but with regards to Voyager specifically I see your point. With how fast this area is moving, I'm also keenly aware of how important it is for the novelty of this work that it gets accepted at this venue.
> > > >
> > > > > Turing completeness in our work specifically relates to the code search space. Unlike prior attempts to automatically discover LLM agents, which often restrict their search space (e.g., focusing solely on prompt optimization), our method defines agents in code. Because the programming language we use (Python) is Turing complete, our search space theoretically allows the discovery of all possible components of agentic systems, such as advanced tool use and RAG in open-sourced codebases like LangChain, and even beyond. As such, our system has more power versus other attempts to automatically create LLM agents. It is true that there are many systems like artificial life systems or even neural networks that are also Turing Complete, and you are right to raise the issue that having this property is not sufficient, for one needs to be able to effectively search this vast search space to produce useful artifacts. But we do show just that. Because it harnesses the powerful intelligence in Foundation Models, Meta-Agent Search marries a Turing Complete search space with the ability to search it well to discover effective solutions.
> > > >
> > > > If your system produced vector in R^n, it would be Turing complete as well. I don't think that property is in any way unique or interesting here and it may be a bit of a red herring, but the exact statements made in the paper aren't technically wrong.
> > > >
> > > > I'm overall satisfied with the paper now. I particularly appreciate how clear the writing is now (something getting increasingly overlooked nowadays). I think this paper would be above the standards of an average paper at ICLR and I've updated my score to reflect that.

---

> > > > > ### Author Response · Authors · 2024-11-26
> > > > > **Thank you!**
> > > > >
> > > > > Dear Reviewer wS7u,
> > > > >
> > > > > Thank you so much for your feedback which has helped us improve the paper and for your strong support in raising your score to 10!
> > > > >
> > > > > We are glad that you noted the writing has become more clear, and we will further improve the discussion regarding Turing completeness in the final paper. In short, we agree with you and will include a comment to the effect that Turing completeness is far from sufficient to create a highly performant system.
> > > > >
> > > > > Thanks again!
> > > > >
> > > > > The authors

---

### Author Response · Authors · 2024-11-24
**General Response (Part 1/3)**

We are deeply grateful to the reviewers for their comprehensive evaluations and thoughtful feedback on our work.

We are encouraged by the reviewers' positive comments, including:
- “Automating agent design has the potential to **accelerate AI development significantly**, and the demonstrated generalization capabilities suggest **wide applicability**.” (MJM9)
- The topic is “relevant” (wS7u) and “thought-inducing” (gtAT). The paper is “timely” (wS7u), “contemporary” (gtAT), and “well-motivated” (955X), “which makes it of interest to a good portion of the community” (gtAT).
- “The introduction of ADAS as a research area is novel” and our proposed algorithm is “well-conceived” and “a creative approach” (MJM9)
- The experiment results show “**consistent improvements over existing methods**” (MJM9) and have “interesting” transferability results (955X).
- The paper is “clearly articulated” and “accessible to readers” (MJM9)

We believe this paper makes an important, helpful contribution that the ML community will benefit from if published. **We appreciate that reviewer MJM9 rated the paper an 8, and reviewer wS7u currently rates it a 5 and is open to increasing their score. But the two other scores are low and likely will prevent publication. We hope all reviewers will keep an open mind to increasing your scores in light of our improvements to the paper in order to improve the chances that the ICLR community will be able to benefit from learning about this work.**

We have addressed all concerns, which has significantly improved the manuscript. In response to your requests, we have produced many new results and improved the paper in the ways you suggested. Here is a quick summary of the major revisions:
- We incorporated a state-of-the-art prompt optimization method (OPRO, Yang et al., 2024) as an additional baseline (Table 1). The results demonstrate that our proposed Meta Agent Search outperforms OPRO across all domains, further reinforcing our argument that defining agents in code and enabling the learning of all components provides significant advantages.

| **Agent Name**              | **Reading Comprehension (F1)** | **Math (%)**  | **Multi-task (%)** | **Science (%)** |
|-----------------------------|-------------------------------|---------------|---------------------|-----------------|
| Prompt Optimization  (Yang et al. 2024)       | 69.1 ± 0.9                   | 30.6 ± 3.2    | 66.4 ± 3.2          | 32.9 ± 3.2      |
| Meta Agent Search (Ours)    | **79.4 ± 0.8**               | **53.4 ± 3.5**| **69.6 ± 3.2**      | **34.6 ± 3.2**  |

- We added experiments running the Meta Agent Search without any initial agent designs (Appendix J). The results show that the discovered agents still outperform hand-crafted baselines across all domains, highlighting that while good initial solutions generally enhance performance, our algorithm can effectively design superior agents from scratch.

| **Agent Name**                     | **Reading Comprehension (F1)** | **Math (%)**  | **Multi-task (%)** | **Science (%)** |
|------------------------------------|-------------------------------|---------------|---------------------|-----------------|
| Meta Agent Search (Empty Initialization) | 73.9 ± 0.9                   | **67.5 ± 3.3**| 68.5 ± 3.3          | 32.7 ± 3.2      |
| Meta Agent Search          | **79.4 ± 0.8**               | 53.4 ± 3.5 | **69.6 ± 3.2**      | **34.6 ± 3.2**  |

- We have radically overhauled the paper’s relation to the topic of AI Safety. Specifically, we had actually done many things to mitigate safety risks in the original work, but we failed to describe them. We will fix that and also explain why we think this work is justified from a safety perspective. See below for more on this important issue that we thank the reviewers for raising.
- We expanded the description of the algorithm in Section 3 and included pseudocode in Appendix I to improve clarity and provide a more detailed explanation of the proposed approach.
- Various other revisions suggested by reviewers were made to enhance precision and clarity throughout the paper.
The revisions have made the paper stronger, and we deeply thank all the reviewers for their help. Below, we respond to shared questions, followed by reviewer-specific responses.

---

> ### Author Response · Authors · 2024-11-24
> **General Response (Part 2/3)**
>
> **Concerns about the safety and the benefit of this work**
>
> Three reviewers (MJM9, 955X, gtAT) expressed concerns about potential risks and safety considerations associated with the automatic generation of agentic systems. Reviewer gtAT worried the approach could be seen as a primitive form of self-replication. Reviewer 955X also inquired about how our work illustrates the potential to benefit humanity.
>
> **Implemented Safety Measures**
>
> We acknowledge the importance of safety in developing systems like ADAS and apologize for not providing detailed explanations of our safety protocols in the original submission. We had, in fact, implemented several safety measures to mitigate potential risks, consistent with safety standards in the published literature (e.g., SWE-Bench, Yang et al., ICML, 2024). We failed to mention these in the original paper (though we should have), which we will fix in the revision. The safety measures are as follows:
> 1. **Containerized Execution**: All generated code is executed within secure containerized environments on remote servers to prevent any unintended side effects or interactions with the host system. This isolation ensures that even if the generated code contains harmful instructions, it cannot affect the underlying system. One could further restrict the sandboxing to proactively block any attempted internet access, but because we have seen zero evidence (see below) that our system is doing anything unsafe with its internet access, and because such access can be important for the science (e.g. to see if it uses web searches in its agentic systems), we currently have left that in place. We will add detailed descriptions of our Containerization practices in the revised paper.
> 2. **Manual Inspection**: We conducted thorough manual reviews of our generated agents to get a sense of whether dangerous agents were being generated that could take potentially harmful actions, such as unauthorized file access, network communications, or execution of system commands. We did this both during preliminary experiments and after our final runs. No such behaviors were observed in our experiments, aligning with our expectations given that the foundation models we used are extensively fine-tuned through Reinforcement Learning from Human Feedback (RLHF) to avoid generating malicious content, and because we do not believe there is any incentive for these agents to take malicious actions. We purposefully avoid using foundation models that have not been “aligned” in this way.
> 3. **Clear Warnings and Guidelines**: We include warnings in our codebase and documentation that will be open-sourced about the potential risks of executing generated code, advising users to employ appropriate safety measures such as sandboxing and strict code review before deploying any generated agents in real-world settings. We will emphasize these guidelines in the revised paper and code release.
>
> **Alignment with Safety Standards in the Literature**
>
> Our safety practices align with those in existing literature where code is generated and executed. Works such as Voyager (Wang et al., TMLR 2024), SWE-Bench (Yang et al., ICML 2024), and FunSearch (Romera-Paredes et al., Nature 2024) have similarly generated and executed code in Turing-complete languages, employing safety measures like sandboxed execution and controlled environments. Some systems like AndroidControl (Wei et al., NeurIPS 2024) and WebAgent (Gur et al., ICLR 2024) even permit broader access, such as modifying local files, or executing system commands, which we do not permit, nor is it our aim in our experiments. By adhering to these established standards, we ensure that our approach is consistent with the community-defined best practices.
>
> **Safety Contributions of ADAS**
>
> Our work offers a pathway to improve the safety of agents. Currently, agent workflows generally fall into two categories:
> 1. **Explicitly Defined Workflows**: These agents rely on explicitly coded logic, such as loops, conditionals, and tool calls (as used in our current algorithm).
> 2. **Autonomous Reasoning-Based Workflows**: Systems like AutoGPT (Richards, 2023) and O1-series models (OpenAI, 2024) depend on the model's reasoning to decide the next step, action, or tool.

---

> > ### Author Response · Authors · 2024-11-24
> > **General Response (Part 3/3)**
> >
> > While both approaches are valid, **explicit workflows provide greater interpretability and controllability**, reducing the risk of malicious behavior by limiting the agent's autonomy. However, prior to our work, the explicit approach required substantial manual effort to design workflows for various applications, which can limit its scalability and performance despite its clear advantages in safety and interpretability. ADAS addresses this limitation by automating the design of workflows in code, making the explicit approach more powerful and inexpensive. This contributes to the creation of agents that are easier to audit, understand, and verify, ensuring safer and more reliable agent behaviors. For example, the visualizations of the generated agents in Fig. 1 and Fig. 3(b) provide clear, auditable representations of their structure and functionality.
> >
> > Additionally, with further integration of safety mechanisms—such as automated checks for generated agents inspired by works like Python package Evalidate, Constitutional AI (Bai et al., 2022), and Thought Cloning’s “Precrime Intervention” (Hu et al., 2023)—ADAS can enhance the applicability and safety of explicitly defined workflows. Embedding these safety checks directly into the agent generation process can prevent the creation of agents with unsafe behaviors, resulting in systems that are both robust and trustworthy.
> >
> > **Potential Benefits to Humanity**
> >
> > AI, especially with advancements in foundation models, has immense potential to benefit humanity in areas such as the economy, health, and beyond (Amodei, 2024). Agentic systems are critical pathways for applying these models, as evidenced by the growing adoption of agent-based techniques by startups and tech companies (Turow, 2024), with reports indicating that nearly half of surveyed tech professionals are already utilizing agents in production (LangChain, 2024). ADAS has the potential to significantly enhance the development of these agents, a point also emphasized by Reviewer MJM9, ultimately contributing to AI’s broader mission of benefiting humanity. Of course, we can only maximize such tremendous potential benefits if we develop the systems in a safe way, which is a point we will state explicitly in our revisions sharing our thoughts and actions on all of these AI safety issues.
> >
> > **Self-replicating AI Systems**
> >
> > It is important to acknowledge that a significant fraction of the AI research community is actively working on self-improving AI systems, such as Clune (AI-GAs, 2019), Fernando et al. (PromptBreeder, ICML 2024), Lu et al. (DiscoPOP, NeurIPS 2024), and Zelikman et al. (STaR, NeurIPS 2022), which have already been published in top-tier venues. It would be an extreme policy to ban publishing all research in this area, especially given its growing prominence. Moreover, the continued development of self-improving AI systems appears inevitable, as this has been a long-standing area of research dating back to foundational works such as Schmidhuber (1987) and each year new work in this direction appears. This makes it essential to ensure that such advancements are approached with a focus on safety. As stated above, our revision will show that our work contributes to safety in this line of research by promoting explicitly defined workflows, interpretability, and controllability in agent designs, while also adopting best practices in AI Safety. This paper aims to raise awareness of this technology’s opportunities while emphasizing the need to create the safest possible versions of these systems. Additionally, it is a benefit to the community to know that this technology is powerful and capable and that we need even more focus on its AI Safety Implications. That is better than banning publications on this topic, which will leave the community unprepared for when other (potentially bad) actors develop it.
> >
> > We acknowledge that our initial submission did not fully convey our deep commitment to safety. In our revised paper, we will emphasize this commitment and actively encourage the development of safety norms for self-improving algorithms. By publishing this work, we hope to inspire further research and discussions on the responsible evolution of self-improving AI technologies, demonstrate the power of such systems, and provide open-sourced code to facilitate safety research in this area.
> >
> > ---
> >
> > By addressing these safety considerations comprehensively and placing the mitigations at the forefront, we aim to reassure the reviewers of our commitment to responsible research and the potential for ADAS to benefit humanity without compromising safety. We will ensure that the revised paper reflects these points in detail.

---

### Meta-Review · Area_Chair_DZhw · 2024-12-20

**Metareview:**

The authors propose a framework to automatically design agentic system through code generation by LLM. The reviewers have very bipolar recommendations (3, 3, 8, 10) and two raise ethical concerns. The negative reviewers' main concerns are safety issues and the lack of state-of-the-art baselines. The paper allows writing agents in any Python code using LLM, which can include malicious code and can create self-replication agents which cannot be controlled. The authors propose several remedies (such as containerized execution, manual inspection, and guidelines). The paper only compare with one other learning algorithm OPRO, missing important recent works [2-3] pursuing in the same direction, as raised by Reviewer 955. If we consider these papers, the novelty of this paper can be less than what's claimed right now.

Nonetheless, I think the strengths of the paper outweigh its weakness, so I will recommend acceptance. Given the preliminary status of this technology and that the paper is applying the algorithm on academic benchmarks rather than building & releasing real-world agents, I think the precaution mentioned in the rebuttal appears to be sufficient to me, if we consider the bar used for deciding acceptance of other agentic papers.

However, I agree with reviewers that **the current writing in the paper on safety is insufficient** and does not fully reflect the points the authors wrote in the rebuttal. Please include and highlight these notes and disclaimers in the revision. As for the concern on weak baselines raised by Reviewer 955X, **please add discussions about [2-3]** and adjust the narrative in the introduction accordingly to timely reflect the current field.

[2] Cheng, C. A., Nie, A., & Swaminathan, A. (2024). Trace is the next autodiff: Generative optimization with rich feedback, execution traces, and llms. arXiv preprint arXiv:2406.16218.

[3] Yuksekgonul, M., Bianchi, F., Boen, J., Liu, S., Huang, Z., Guestrin, C., & Zou, J. (2024). TextGrad: Automatic" Differentiation" via Text. arXiv preprint arXiv:2406.07496.

**Additional Comments On Reviewer Discussion:**

Reviewer wS7u raises concerns on novelty, overemphasis on Turing completeness, and clarity on the algorithm. The responses fully address the reviewer's concerns as reflected in the increased score.

Reviewer MJM9 raises issues about safety concern, baseline choices. and scalability. The responses address all concerns.

Reviewer 955X raises concerns on Turing completeness, missing important disclaimer (the meta optimizers is different for different tasks.), weak baselines, writing clarity, and missing ablation experiments. While the rebuttal addresses most of the concerns, the concern on weak baselines remain. I tend to side with the reviewer on this point. While [1] focuses on prompt+connectivity optimization, [2] demonstrates optimizing agent code or code+prompt jointly, and [3] can optimize any texts and therefore can be used to optimize code (which is also text). So I think the authors' claim "[2] and [3] also primarily optimize prompts" is inaccurate and a direct comparison is possible (both have open-sourced libraries). Comparison with [2] and [3] are needed in this paper given how fast this field grows now (both were arxived in June and [2] was published in NeurIPS 2024 already); [2] especially demonstrates in-place agent code optimization as its key feature, which would affect the novelty claim of this paper. Nonetheless, I agree with the authors that it was nearly impossible to conduct new experiments near the end of the rebuttal.

Reviewer gtAT raises concerns on safety, writing clarity, and computational cost. I agree with the reviewer on the computational cost. Though I don't think this would be a reason for rejection, I would suggest **providing computational cost of the proposed method and the other baselines used in the experiments,** e.g., in terms of the tokens used (training & inference; e.g. does the learned agent use more computes/tokens to perform better?)

---

### Decision · Program_Chairs · 2025-01-22

Accept (Poster)